# ◷ VERSICODE: TOWARDS VERSION-CONTROLLABLE CODE GENERATION

## ABSTRACT

Large Language Models (LLMs) have made tremendous strides in code generation, but existing research fails to account for the dynamic nature of software development, marked by frequent library updates. This gap significantly limits LLMs' deployment in realistic settings. In this paper, we propose two novel tasks aimed at bridging this gap: version-specific code completion (VSCC) and version-aware code migration (VACM). In conjunction, we introduce VersiCode, a comprehensive Python dataset specifically designed to evaluate LLMs on these two tasks, together with a novel evaluation metric, Critical Diff Check (CDC@1), which assesses code generation against evolving API requirements. We conduct an extensive evaluation on VersiCode, which reveals that version-controllable code generation is indeed a significant challenge, even for GPT-4o and other strong frontier models. We believe the novel tasks, dataset and metric open up a new, important research direction that will further enhance LLMs' real-world applicability. The code and resources can be found at `https://anonymous.4open.science/VersiCode-B0F6`.

## 1 INTRODUCTION

Large Language Models (LLMs), including OpenAI's GPT series (OpenAI, 2023a;b; 2024) and specialized variants such as CodeLLaMA (Rozière et al., 2023), have demonstrated significant advancements in code generation tasks. Typically evaluated using benchmarks such as HumanEval (Chen et al., 2021) and MBPP (Austin et al., 2021), these models are measured on tasks that assume code generation is a *static* activity. However, the reality of software development is inherently dynamic, characterized by frequent updates to software libraries, which necessitate adjustments to API interfaces. This evolving landscape raises crucial challenges for LLMs, particularly their ability to generate code that is functional for different, specific library versions. This dynamic nature of software development leads us to ask the following questions:

- How reliably can LLMs generate code compatible with specific library versions?
- How effectively can LLMs adapt code for API changes across library versions?

Existing benchmarks (Jiang et al., 2024; Sun et al., 2024; Luo et al., 2024b), which are oblivious to version-specific dynamics, do not address these challenges. They fall short of simulating the continuous version management activities undertaken by developers who ensure the software remains functional across updates. The static nature of existing benchmarks represents a significant barrier to the practical deployment of LLMs in professional environments, where handling version-specific dependencies is critical (Zhang et al., 2020; 2021; Dilhara et al., 2021; Liu et al., 2021; Wang et al., 2020; Vadlamani et al., 2021; Haryono et al., 2021).

To bridge this gap, we propose two novel tasks aimed at evaluating LLMs' version-controllable code generation capabilities, namely version-specific code completion (VSCC) and version-aware code migration (VACM). These tasks are crafted to mimic real-world software development scenarios, motivated in Figure 1, requiring models to generate code that not only is syntactically correct but also adheres to version-specific API contracts (Zhang et al., 2020; 2021; Dilhara et al., 2021; Liu et al., 2021; Wang et al., 2020; Vadlamani et al., 2021; Haryono et al., 2021). Moreover, we introduce VersiCode, the first dataset specifically designed for these two tasks. VersiCode includes data spanning over 300 Python libraries and more than 2,000 versions across 9 years. It has undergone a careful curation process to ensure high quality. Thus, VersiCode provides a comprehensive and robust testbed

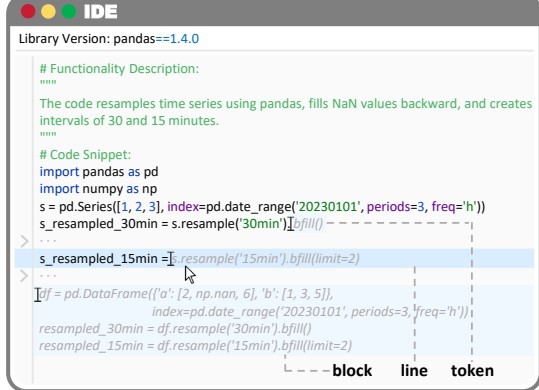

Figure 1: Two motivating scenarios for version-controllable code generation: (left) Interacting with LLMs in a browser, where slight query changes lead to incorrect answers, and (right) Programming in an IDE, explicitly specifying the version of dependency libraries.

for assessing LLMs under realistic conditions. Furthermore, we propose a new evaluation metric, CDC (Critical Diff Check), which enhances traditional code similarity metrics by incorporating considerations for API usage, parameter handling, and deprecated features management. This metric offers a more granular assessment of a model's ability to navigate the complexities of evolving software libraries.

Our extensive testing of strong frontier models like GPT-4o and LLaMA3 (Meta LlaMa team, 2024) reveals significant challenges in version-aware code generation tasks. We uncover that (1) LLMs often retain outdated programming knowledge, particularly concerning version-specific information. (2) Conventional metrics used for evaluating code generation do not effectively capture the nuances of version sensitivity. (3) While leveraging context from various library versions can be beneficial, its utility can be limited. Guided by these insights, we suggest strategies, such as targeted pretraining, continual learning, and refined evaluation methods, for improving LLMs' version-controlled code generation capabilities.

Our contributions are summarized as follows:

- We propose two novel and important yet under-explored tasks in code generation, namely version-specific code completion and version-aware code migration.
- We introduce VersiCode, a comprehensive, well-documented and *versioned* dataset, accompanied by a subset annotated with executable test cases.
- We introduce Critical Diff Check, a new metric that extends traditional code similarity metrics by checking syntactic validity, API usage, parameter matching, the use of 'with' statements, and correct keyword arguments in the generated code, providing a more detailed evaluation of version-specific code generation.
- Our thorough experiments provide valuable insights and directions for future research in this critical area of software development.

## 2 VERSION-CONTROLLABLE CODE GENERATION

VersiCode is a large-scale code generation benchmark dataset focusing on evolving library dependencies. We curated our dataset by initially selecting popular Python repositories from GitHub, confirmed by their star ratings, and ensured they were permissively licensed. For each library, we compiled data from three main sources: (1) Library Source Code, extracting all pip-installable versions and official API usage examples from docstrings; (2) Downstream Application Code, sourcing from top-tier research papers spanning ten years to capture evolving libraries; (3) Stack Overflow, retrieving FAQs that mention specific library versions. We present the dataset statistics, construction process and examples in detail in Appendix 2.

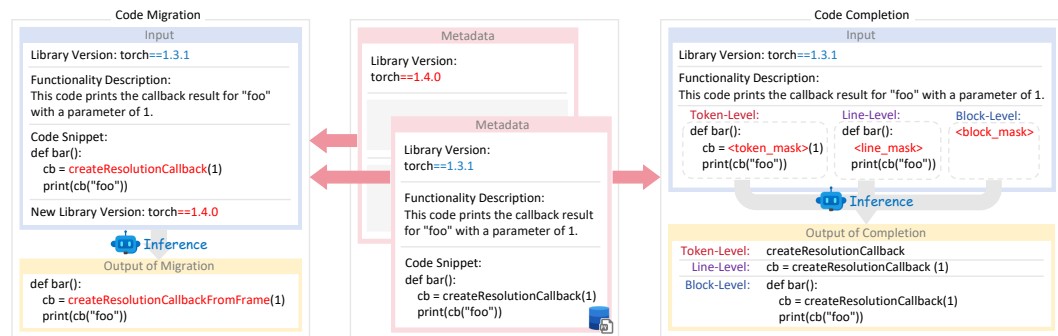

Figure 2: The post-processing pipeline transforms metadata into specific tasks and the running example per task: (left) Leveraging pairs of metadata that share the same functionality but different library versions to construct block-level code migration instances; (right) Utilizing each metadata sample, masking version-sensitive content to create multi-granularity code completion instances.

As shown in Figure 2, we define a *meta-instance* as $m = [l; v; d; c] \in \mathcal{M}$, where $l$, $v$, $d$, and $c$ represent the library name, version, functionality description, and code snippet, respectively. Consider an API $a$ added to library $l$ in version $v_s$ and deprecated in version $v_e$, and is active in the intermediate version $v_m$ where $s \leq m \leq e$. We refer to the interval $[s, e]$ as the *lifecycle* of $a$. To analyze model performance in detail, we assess how up-to-date each LLM is concerning newly added or deprecated APIs per version. We compare the source code between any two consecutive versions of each library to detect changes in API or method names. Based on the detection results, we label the source code as follows: "addition" indicates an API newly added in the current version and still applicable in subsequent versions; "deprecation" indicates the current version is the last usable version for the API; and "general" indicates the API usage method is inherited from the previous version.

We introduce the two novel version-controllable code generation tasks below.

**Version-Specific Code Completion (VSCC)**: Given a meta-instance $m_i$, the input is $x = [l_i; v_i; d_i; c'_i]$, where $c'_i$ is the code snippet $c_i$ with selective masking, replacing the library- and version-sensitive contents with a special token. Depending on the length of the masked contents, the special token is defined as "[token-mask]", "[line-mask]", or "[block-mask]", reflecting code completion on different granularity levels. The output $y$ is the masked content, typically containing function names or variables.

**Version-Aware Code Migration (VACM)**: Given a pair of meta-instances $(m_i, m_j | l_i = l_j, d_i = d_j, v_i \neq v_j)$, the input $x = [l_i; v_i; d_i; c_i; v_j]$, and the output $y = c_j$. Note that version editing may require refactoring of the code structure, making it difficult to format as detailed as in token-level or line-level completion. Additionally, depending on the numerical relationship between $v_i$ and $v_j$, various scenarios arise, such as editing from an old version to a new version, or vice versa. Data statistics are detailed in Appendix B

## 3 TOKEN-LEVEL VERSION-SPECIFIC CODE COMPLETION

In code generation that targets a specific version of a third-party library, the version-related changes usually involve updates to identifiers, such as the addition, removal, or renaming of classes, functions, and parameters. The token-level code completion task for a specified version, predicting the evolving identifiers identified in real code, is a fundamental and direct way to evaluate LLMs to generate code for specific versions. We begin our research by addressing the following three research questions: (1) How well do LLMs perform on code completion tasks that involve version-specific library usage compared to other benchmarks like HumanEval and MBPP? (2) How do LLMs handle new, deprecated, and intermediate versions of libraries in code completion tasks? (3) How does the performance of LLMs in code completion change over time with different library versions?

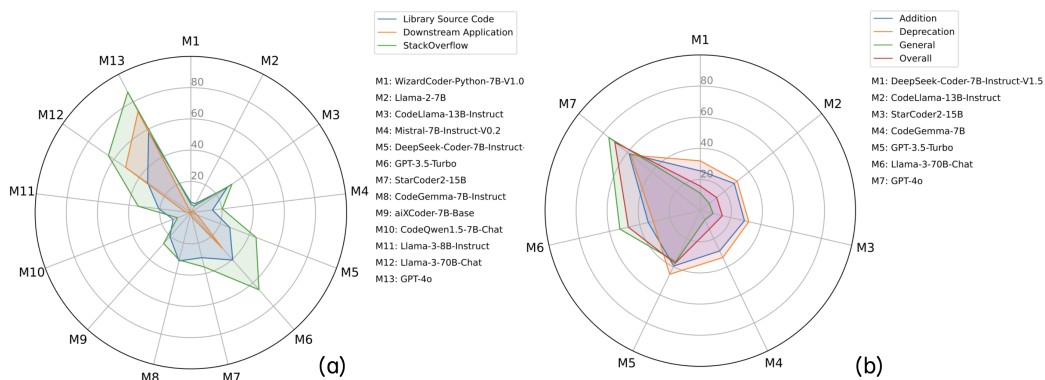

Figure 3: The *EM@1* results for token-level code completion from VersiCode: Performance grouped by data sources, and (b) Performance grouped by API lifecycle.

## 3.1 Experiment Setup

**Models**: We benchmarked VersiCode against popular open-domain LLMs and dedicated code-LLMs, including variant families such as GPT (OpenAI, 2023a;b; 2024), LLaMa (Touvron et al., 2023), Mistral (Jiang et al., 2023), CodeLLaMa (Rozière et al., 2023), CodeQwen (Bai et al., 2023), CodeGemma (CodeGemma Team et al., 2024), StarCoder (Lozhkov et al., 2024), Deepseek-Coder (Guo et al., 2024), and WizardCoder (Luo et al., 2024c). For smaller open-source models (e.g., <20B parameters), we downloaded them from HuggingFace [1] and deployed them locally for inference. For larger models, such as LLaMa3 70B (Meta LlaMa team, 2024) and GPT-4o (OpenAI, 2024), we used their online APIs [2] [3] for inference.

**Data Preparation**: Each instance in VersiCode is tagged with its data source (library source code, downstream applications, or Stack Overflow), feature type (addition, deprecation, or general), and release time, allowing for more detailed performance analysis. We randomly selected 2,000 instances for token-level code completion. (see Appendix A.3).

**Baseline Dataset**: To assess the difficulty of VersiCode, we compared it with two well-known code generation datasets, HumanEval (Liu et al., 2023) and MBPP (Jiang et al., 2024), and observed the overall performance of models. HumanEval (Liu et al., 2023) measures functional correctness in synthesizing programs from docstrings with 164 original problems, resembling simple software interview questions. MBPP (Austin et al., 2021), with about 1,000 crowd-sourced Python problems for entry-level programmers, covers programming fundamentals and standard library functionality, including task descriptions, code solutions, and three automated test cases for each problem. We also collected the evaluation results for their upgraded versions HumanEval+ (Liu et al., 2023) and MBPP+ (Liu et al., 2023). Please refer to Appendix D.1 for details.

**Evaluation Metrics**: We use **EM@$k$** for token-level generation: For this metric, we generate $n \geq k$ samples per instance (with $n = 100$ and $k \in \{1, 3, 10\}$ for our experiments). We count the number of correct samples $c \leq n$ judged by exact matching. @$k$ is defined as the average performance over the task, calculated as $\mathbb{E}\left[1 - \frac{\binom{n-c}{k}}{\binom{n}{k}}\right]$, which is the same with Pass@$k$ (Chen et al., 2021).

## 3.2 Results and Analysis

However, a substantial performance gap of at least 15 points remains when compared to HumanEval and MBPP (detailed in Appendix D.1). This indicates that state-of-the-art LLMs still struggle to deliver satisfactory results, even for the simplest token-level completion tasks.

---

[1] https://huggingface.co/models
[2] https://together.ai
[3] https://openai.com/

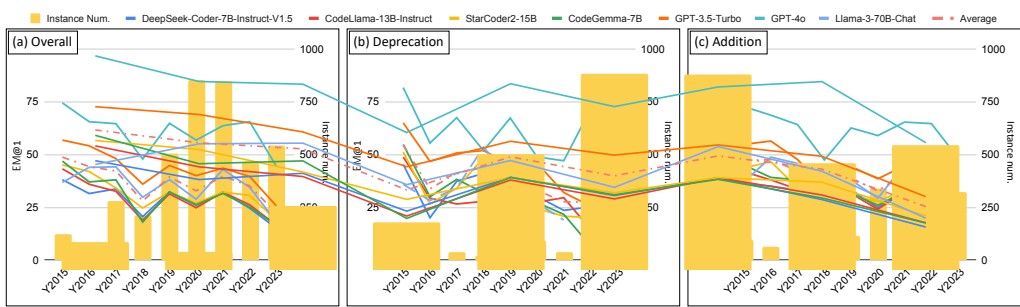

Figure 4: The *EM@1* performance for token-level code completion, grouped by year (2015-2023), with a histogram of data distribution for each year.

**Differences in LLM performance across different data sources.** Figure 3-a presents the EM@1 results for token-level code completion on VersiCode, categorized by data sources. Among the three data sources, most models perform significantly better on Stack Overflow, especially compared to handling source code from downstream applications. This discrepancy may be attributed to the greater diversity found in downstream applications, which demands a more robust capability to address varied challenges. This may also indicate that Stack Overflow is heavily represented in the pre-training data of LLMs, increasing the likelihood of data leakage. GPT-4o (M13) and LLaMA3-70B (M12) stand out as outliers, increasing the likelihood of models memorizing specific content, which may excel in handling downstream applications. Full numeric results are provided in Appendix D.1.

**Challenges in casual intermediate library versions.** We present the token-level EM@1 results for the token-level code completion task, categorized by lifespan features: addition (in blue), deprecation (in orange), and general (referring to intermediate versions; in green), as shown in Figure 3-b. Most models perform well in cases of addition and deprecation, likely because newly added or deprecated APIs are often emphasized in documentation and by the community. However, most models struggle with reasoning and adapting to intermediate versions. As shown in Figure 3-a, models like LLaMA3-70B excel in downstream applications and handle intermediate versions more effectively, likely due to the diversity of use cases they encounter.

**The programming knowledge of LLMs, particularly regarding version-specific information, is surprisingly outdated.** Figure 4 presents the EM@1 performance for token-level code completion, grouped by year from 2015 to 2023, along with a histogram showing the data distribution for each year. To ensure precise timestamps and minimize noise, we only used instances collected from library source code. As shown in Figure 4-a, there is a clear trend: model performance declines as the release time becomes more recent. This is counter-intuitive compared to temporal knowledge question answering (Zhao et al., 2024), where performance initially increases before declining. We further filtered for "deprecation" (Figure 4-b) and "addition" (Figure 4-c) to identify version-sensitive cases. Although data sparsity reduces confidence in the results, both cases show a clear downward trend over time This suggests that LLMs have outdated programming knowledge, highlighting the need for rapid adaptation to newer libraries and APIs.

## 4 FROM TOKEN-LEVEL TO LINE- AND BLOCK-LEVEL COMPLETION

When utilizing third-party code library APIs, LLMs should handle not only API name generation but also parameter preparation and contextual code integration. In this section, we extend the task to line-level (completing a single line) and block-level (completing multiple lines) code generation. This expanded scope presents new challenges for both the model's capabilities and the evaluation methodologies. (1) How does increasing complexity in line- or block-level code completion affect the LLMs to handle API usage and parameters? (2) How does having more context (like import statements and specified library version) improve the accuracy of line- and block-level code generation? (3) Which evaluation metrics best capture the accuracy of line- and block-level code generation, and which is most reliable?

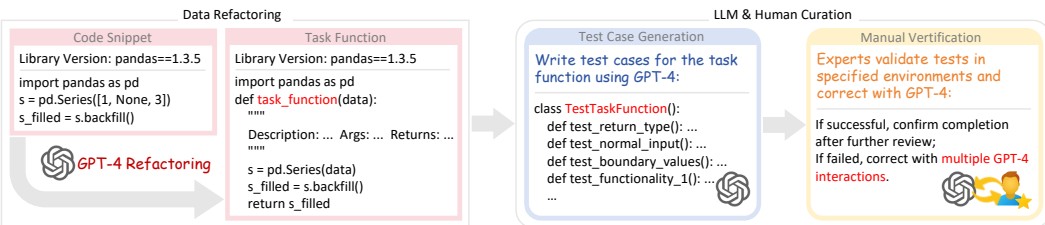

Figure 5: The process of executable code assessment, which includes data refactoring, test case generation, and validation. Starting from code snippets collected from real code involving specific API calls for a given library version, GPT-4 is employed to refactor the code into a task function. The large language model is then prompted to generate test cases from various perspectives (See Appendix F for a running example of instances and test cases.) Each generated test case is verified by experts, and the correctness is ensured by running the code in a specified environment. If issues arise, they are corrected through multiple iterations with GPT-4.

## 4.1 EXPERIMENT SETUP

**Models**: We selected GPT-4o, GPT-3.5, and LLaMA3 70B, the three models that perform best on token-level code completion, to conduct experiments on line-level or block-level code completion.

**Data Preparation**: We sample a subset from VersiCode for dynamic code analysis with executable test cases from library source code, focusing on code snippets with complete context (e.g., import statements). GPT-4 was used to refactor the snippets into task functions, followed by test case generation and validation in a version-specific environment. All of the test cases have been manually verified to ensure their correctness. The code completion tasks are categorized into token, line, and block levels. The test cases include return type, normal input, boundary values, and functionality checks (see Appendix A.3 for details).

**Metrics**: We use the following evaluation metrics for each task granularity: (1) **Pass@**$k$ for token-level generation (Chen et al., 2021): For this metric, we generate $n \geq k$ samples per instance (with $n = 6$ and $k = 1$ to compare different metrics). We count the number of correct samples $c \leq n$ judged by executable testing. (2) **Identifier Sequence Match (ISM@**$k$**)** and **Prefix Match (PM@**$k$**)** for line-level generation (Agrawal et al., 2023): These metrics measure how closely the generated sequences match the ground truth. For block-level generation, we adopt the average performance over lines. Following the setup in Agrawal et al. (Agrawal et al., 2023), we generate $n = 6$ independent samples per instance. (3) **Exact Match (EM@**$k$**)**: We use regular expression matching to determine whether the specified API is used in the code generated by the model and the formula for calculating the EM@k score is the same as the formula for calculating the Pass@k score ($n = 6$ and $k = 1$). (4) **Critical Diff Check (CDC@**$k$**)**: Unlike traditional code similarity calculations, CDC focuses on the differences between the code generated by the model and the reference answer. CDC extends the EM metric by adding four additional rules: checking whether the generated code is syntactically valid; identifying the line in the generated code where the specified API is used and determining if the number of parameters in the function call is the same; if the answer uses a with statement, checking whether the generated code also uses a with statement; and if the answer uses keyword arguments, verifying whether the generated code uses the same keyword arguments. Please refer to Appendix E for detailed examples, effectiveness analysis, and ablation study, conducted to validate CDC.

## 4.2 RESULTS AND ANALYSIS

**Less context leads to more errors in code generation.** When models have more context, like import statements, their performance improves significantly. For example, as shown in Table 1, GPT-4o at the token level achieves a Pass@1 score of 65.97 with imports, but this drops to 44.54 without imports. This pattern is consistent across all models and granularity levels (i.e., token, line, and block), as shown in Figure 6. When models lack important context, such as external libraries or other dependencies, they struggle to generate accurate code, which leads to more errors. So, giving models more information upfront is crucial for better results.

**Models show limited sensitivity to version-specified instructions.** As shown in Table 1, at the token level, models like GPT-4o perform slightly better when provided with version information

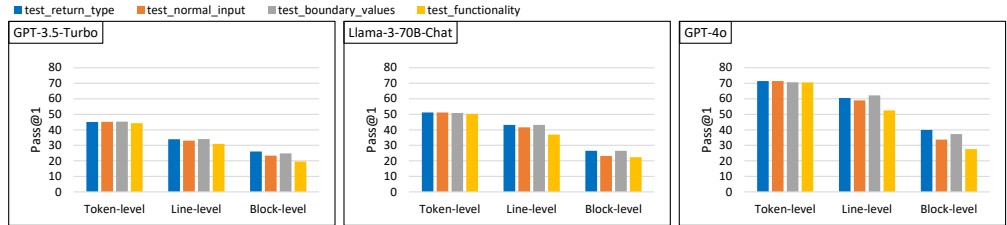

Figure 6: The *Pass@1* performance of different models across various granularities and test case types.

| Model | Token | | | Line | | | | | Block | | | | |
|-------|-------|-------|-------|-------|-------|-------|-------|-------|-------|-------|-------|-------|-------|
| | Pass@1 | EM@1 | CDC@1 | Pass@1 | EM@1 | ISM@1 | PM@1 | CDC@1 | Pass@1 | EM@1 | ISM@1 | PM@1 | CDC@1 |
| *w/ import; w/ version* | | | | | | | | | | | | | |
| GPT-3.5-Turbo | 44.68 | 49.16 | 49.16 | 30.67 | 59.38 | 47.44 | 40.01 | 27.59 | 14.85 | 54.90 | 78.15 | 47.04 | 26.61 |
| Llama-3-70B-Chat | 50.28 | 54.48 | 54.48 | 36.27 | 63.87 | 49.58 | 45.50 | 29.27 | 15.41 | 59.10 | 79.10 | 48.02 | 27.73 |
| GPT-4o | 71.15 | 76.33 | 76.33 | 51.82 | 78.99 | 57.98 | 59.49 | 41.04 | 22.55 | 59.24 | 76.68 | 59.43 | 31.65 |
| *w/o import; w/ version* | | | | | | | | | | | | | |
| GPT-3.5-Turbo | 22.27 | 23.95 | 23.95 | 17.51 | 33.61 | 29.79 | 25.80 | 16.11 | 3.36 | 25.77 | 61.57 | 35.43 | 10.50 |
| Llama-3-70B-Chat | 25.49 | 27.45 | 27.45 | 19.61 | 38.38 | 30.73 | 26.48 | 15.27 | 4.62 | 34.03 | 68.12 | 43.91 | 10.78 |
| GPT-4o | 52.80 | 56.86 | 56.86 | 35.71 | 61.90 | 46.22 | 40.28 | 28.29 | 7.28 | 45.52 | 74.85 | 51.91 | 18.07 |
| *w/ import; w/o version* | | | | | | | | | | | | | |
| GPT-3.5-Turbo | 44.54 | 49.02 | 49.02 | 31.79 | 60.5 | 48.28 | 42.27 | 27.45 | 15.97 | 57.14 | 78.42 | 48.21 | 28.71 |
| Llama-3-70B-Chat | 49.44 | 54.06 | 54.06 | 35.01 | 62.89 | 61.34 | 62.93 | 29.13 | 13.31 | 59.38 | 77.96 | 56.08 | 26.33 |
| GPT-4o | 71.01 | 77.03 | 77.03 | 51.12 | 77.31 | 50.05 | 46.45 | 40.90 | 24.79 | 64.71 | 79.59 | 49.38 | 33.33 |
| *w/o import; w/o version* | | | | | | | | | | | | | |
| GPT-3.5-Turbo | 22.41 | 24.23 | 24.43 | 17.65 | 36.97 | 31.93 | 27.29 | 17.65 | 3.92 | 27.31 | 62.34 | 36.62 | 10.64 |
| Llama-3-70B-Chat | 25.77 | 29.13 | 29.13 | 19.47 | 40.20 | 36.61 | 28.95 | 16.95 | 4.06 | 33.47 | 66.52 | 42.52 | 12.61 |
| GPT-4o | 49.72 | 54.90 | 54.90 | 35.15 | 61.76 | 47.90 | 42.41 | 29.27 | 5.60 | 46.36 | 74.95 | 50.15 | 17.93 |
| *Pearson Correlation Coefficient with Pass@1* | | | | | | | | | | | | | |
| PCC | - | 0.9995 | **0.9995** | - | 0.9810 | 0.8314 | 0.8196 | **0.9917** | - | 0.8974 | 0.7912 | 0.6547 | **0.9626** |

Table 1: The performance of different models across various granularities *(Token, Line, Block)*. *Pass@1* refers to dynamic analysis metrics, while green-colored metrics *(EM, ISM, PM)* correspond to static analysis based on string matching. The blue-colored metric *(CDC)* represents a newly proposed metric. The configurations labeled as *"w/o version"* indicate that the prompt does not specify the version of the third-party code libraries, while *"w/o import"* refers to prompts where the provided code context lacks import statements, meaning the model must generate code based entirely on user intent. The Pearson correlation coefficient is computed for each metric's results against Pass@1 within each granularity.

(52.80 with version v.s. 49.72 without version). However, this advantage diminishes at the line and block levels, where the results become inconsistent. This suggests that while version details can be helpful for short code snippets, they don't significantly impact the model's performance for more extended or complex code. This likely indicates that models are not trained to prioritize or heavily rely on version-specific instructions.

**The CDC@1 metric closely aligns with Pass@1 scores, making it a strong proxy for dynamic code analysis.** As shown in Table 1, at the block level, the Pearson Correlation Coefficient (PCC) between CDC@1 and Pass@1 is 0.9995, indicating a strong correlation. Even though EM@1 has a high correlation with Pass@1 at the token level (PCC = 0.9995), EM@1 becomes less aligned at the block level (PCC = 0.8974). Additionally, the absolute differences between CDC@1 and Pass@1 values are generally smaller compared to other static metrics like EM@1, making CDC a potentially more reliable alternative for assessing code generation accuracy.

## 5 FROM CODE COMPLETION TO CODE MIGRATION

In addition to generating code for specific third-party library versions, another common challenge is maintaining user projects when these libraries are upgraded or rolled back. We address version-aware code migration by exploring three key questions: (1) How well can LLMs handle migrating code across different versions, compared to generating code for a specific version? (2) What impact do major and minor version changes in third-party libraries have on code migration? (3) How do forward migrations (from older to newer versions) compare to reverse migrations (from newer to older versions) in terms of trends and challenges?

| Model | Various Version Type | | | | | | | | Various Releasing Time | | | |
| --- | --- | --- | --- | --- | --- | --- | --- | --- | --- | --- | --- | --- |
| | Major→Major | | Major→Minor | | Minor→Major | | Minor→Minor | | Old ↦ New | | New ↦ Old | |
| | CDC@1 | CDC@3 | CDC@1 | CDC@3 | CDC@1 | CDC@3 | CDC@1 | CDC@3 | CDC@1 | CDC@3 | CDC@1 | CDC@3 |
| DeepSeek-Coder-7b-instruct-v1.5 | 4.08 | 8.00 | 5.50 | 11.47 | 7.00 | 14.38 | 8.08 | 17.23 | 14.5 | 28.89 | 9.16 | 21.18 |
| CodeLLaMA-13b-Instruct-hf | 2.33 | 5.60 | 4.08 | 9.68 | 7.58 | 16.58 | 7.00 | 15.80 | 8.14 | 18.4 | 9.92 | 22.82 |
| StarCoder2-15b | 2.25 | 4.60 | 2.58 | 6.40 | 5.00 | 11.88 | 4.83 | 11.75 | 6.49 | 14.85 | 5.22 | 13.51 |
| CodeGemma-7B | 1.00 | 2.80 | 0.25 | 0.75 | 0.00 | 0.00 | 0.50 | 1.40 | 0.13 | 0.38 | 0.38 | 1.15 |
| GPT-3.5-turbo | 5.00 | 6.50 | 9.67 | 14.90 | 19.00 | 25.23 | 19.42 | 25.40 | 22.14 | 29.73 | 19.47 | 32.21 |
| LLaMA-3-70b-chat | 12.92 | 14.20 | 13.42 | 16.20 | 13.08 | 15.50 | 16.33 | 19.82 | 15.27 | 19.73 | 19.97 | 30.76 |
| GPT-4o | 23.67 | 25.95 | 35.25 | 38.40 | 42.08 | 47.53 | 37.83 | 47.40 | 43.00 | 48.02 | 38.42 | 47.37 |

Table 2: The performance of various models in different code migration scenarios. The arrow "↦" indicates the direction of migration, where "Major" denotes the major release (e.g., Torch v2.0.0 and v2.4.0), and "Minor" denotes the minor release (e.g., Torch v2.1.3 and v2.3.4). Therefore, the migration could be categorized as (1)"{x}↦Major", crossing any major release, like from v2.0.0 to v2.4.0; (2) "{x}↦Minor", migrating to a version before the next major release, like from v2.0.0 to v2.0.3. The "Old ↦ New" scenario simulates upgrading from an old version to a new version, while "New ↦ Old" represents the maintenance of historical code. The performance of different models in these scenarios is measured using the CDC metrics (CDC@1 and CDC@3), reflecting their adaptability to various code migration tasks.

## 5.1 EXPERIMENT SETUP

**Models**: Based on the token-level code completion experimental results in Section 3, we selected the most outstanding performers from each model series for the experiments in this section.

**Data Preparation**: For code migration, we utilize a subset of VersiCode, in which instances are constructed based on differences between source and target code versions, covering both updates to newer versions and downgrades. Versions were categorized by patterns (e.g., major vs. minor) to capture different migration scenarios. (Detailed in Appendix A.3)

**Metrics**: Code migration is similar to block-level tasks in code completion. We use the same evaluation metric as for block-level: CDC@$k$ ($n = 6$, $k \in \{1, 3\}$).

## 5.2 RESULTS AND ANALYSIS

**Model performance across version migrations.** Different models display varying degrees of adaptability when transitioning between major and minor software versions, with some showing exceptional robustness in Table 2. The table categorizes version migrations into four types: Major-to-Major, Major-to-Minor, Minor-to-Major, and Minor-to-Minor. Notably, most models did not exhibit a significant pattern across different migration scenarios, likely due to their limited awareness of version-specific API knowledge. Among the scenarios, "Minor↦Minor" intuitively represents the simplest case (requiring the least code modification). Interestingly, GPT-4o's performance is particularly remarkable in the "Minor↦Major" scenario, where it achieves the highest effectiveness.

**Adaptability in code migration based on release timing.** Backward and forward compatibility testing reveals a spectrum of model resilience under different temporal migration scenarios. The evaluation is split into two releasing time directions: Old-to-New and New-to-Old, shown in Table 2. Generally, models perform better when adapting to newer versions from older ones, with GPT-4o standing out for its high scores in both directions. However, the drop in performance when handling older versions after training on newer releases highlights challenges in maintaining backward compatibility, a critical aspect for long-term usability and integration stability in evolving tech environments.

**The context code in another version is still helpful, but its benefits are limited.** The comparison between block-level code completion and block-level code migration is shown in Table 2 and Table 1, reorganized in Appendix D.3, especially Table 9. There is a significant improvement across most models, except for LLaMA3-70B and GPT-4o. When provided with code in another version as context (i.e. in the code migration task), these models can generate correct code with a much higher success rate. However, a bottleneck is more evident in LLaMA3-70B and GPT-4o, where the code context hinders their performance than code completion.

## 6 DISCUSSION

**How can we enhance pre-training for new code-LLMs?** Figure 4 demonstrates a notable decline in the performance of all models over time. This deterioration is likely attributable to two primary factors: (1) the use of outdated pre-training data, which causes older versions of code to predominate the training set, and (2) the backward compatibility of APIs, which results in a higher prevalence of use cases and examples about older versions of these APIs (Lamothe et al., 2022). To mitigate this issue and improve the models' capabilities with newer libraries, we suggest increasing the representation of new-version codebases within the training data. This adjustment aims to enhance the proficiency in utilizing contemporary libraries effectively (Zhao et al., 2024; Shao et al., 2024). Besides, based on the results in Section 4.2, current LLMs show limited use of version information in code generation. To address this, we propose enhancing pre-training by incorporating version-tagged code samples and metadata to help models better differentiate between API versions.

**How can we address the challenge of evolving libraries in LLMs?** Generating block-level or repository-level code (Luo et al., 2024a) requires LLMs to understand user demands and library dependencies. Addressing this challenge involves continually training the model with new libraries using continual learning techniques (Jiang et al., 2024). These techniques enable the model to adapt to changing libraries without forgetting previously learned information. Examples include memory-based methods and various continual learning strategies (Wu et al., 2024; Yadav et al., 2023; Wu et al., 2022). Additionally, developing benchmark datasets that are continuously and automatically curated and maintained is crucial for evaluating the performance of models with new libraries (Jang et al., 2022). Enriching the taxonomy (Jiao et al., 2023) and maintaining datasets for evolving libraries (Lamothe et al., 2022) is also vital (Jiao et al., 2023). Multi-agent systems can be employed for this purpose. Aligning development and evaluation efforts will enhance the ability of LLMs in code understanding and generation capabilities, to remain effective as libraries evolve.

**Can we address version-controllable code generation with retrieval-augmented generation?** Retrieval-augmented generation (RAG) approaches typically involve two crucial components: retrieval and in-context generation (Gao et al., 2023). The following challenges need to be addressed in order for RAG to be effectively applied to this problem. From the **retrieval** perspective: (1) It may be difficult to disambiguate version-related queries, as embeddings for version strings like "torch 2.1.3" and "torch 1.3.2" can be very similar (Singh & Strouse, 2024). This similarity makes it hard for retrievers to differentiate between specific features and capabilities associated with each version. ( 2) Version information of code snippets is rarely explicitly mentioned within the code itself and may instead appear in separate configuration files like "requirements.txt". This separation necessitates a more sophisticated retrieval approach, where the model must integrate information from multiple sources to accurately understand version dependencies. From the perspective of **in-context generation**: Table 9 shows that even non-matching version contexts (i.e., code migration) can help smaller models generate grammatically correct code. This observation suggests potential for dedicated RAG approaches (Jiang et al., 2024), though the benefits are limited and retrieval noise may reduce effectiveness.

**What are the effective methods for evaluating the capabilities of LLMs in generating version-controllable code?** Both static analysis (Agrawal et al., 2023), which reviews code without executing it, and dynamic analysis (Zhuo et al., 2024), which tests the code by running it, are vital for software development. However, evaluating LLMs for version-controllable code generation presents unique challenges. (1) Dynamic analysis is complicated by API calls that rely on specific code contexts, making it difficult and costly to create standalone tests (Zhuo et al., 2024). Additionally, using LLM-generated code as test cases introduces further complexity in managing test quality. Especially, VersiCode, which includes 300 packages and over 2,000 versions in the raw dataset, requires detailed setups for each testing environment and managing various dependencies, complicating the practical deployment of solutions. (2) Meanwhile, static analysis uses metrics like ISM (Agrawal et al., 2023) and PM (Agrawal et al., 2023) for broad coverage but may miss critical details such as indentation and parameter positioning in API-related code, refer to Table 1 and Appendix E. These omissions suggest that traditional static metrics are not entirely suitable for assessing version-controllable code generation. Evaluating the effectiveness of these metrics is crucial. Our study initiates the exploration of more reliable methods; however, extensive research, including approaches like code slicing (Du et al., 2024), is essential to advance our evaluation techniques.

## 7 RELATED WORK

**Code Generation Models**: Recent advancements in code language models (Guo et al., 2024; CodeGemma Team et al., 2024; Bai et al., 2023; Rozière et al., 2023; Sun et al., 2024), driven by sophisticated NLP techniques (Jiang et al., 2024) and extensive code repositories (Hu et al., 2023), have resulted in substantial breakthroughs. Transformer-based large language models (Luo et al., 2024c; Rozière et al., 2023; Guo et al., 2024; Lozhkov et al., 2024; Bai et al., 2023; Gunasekar et al., 2023; Li et al., 2023) have demonstrated exceptional capabilities in generating syntactically correct and semantically meaningful code from natural language descriptions. Additionally, research efforts that integrate multi-modal data (OpenAI, 2023b; 2024; Meta LlaMa team, 2024), including both code and accompanying documentation (Hu et al., 2023), have significantly improved model accuracy. While in real-world software engineering,

**Code Generation Datasets**: The code generation (Jiang et al., 2024; Sun et al., 2024; Luo et al., 2024b) includes tasks for both code completion and code editing, ensuring comprehensive coverage of programming scenarios. Code completion (Yao et al., 2018; Yin et al., 2018; Feng et al., 2020; Chen et al., 2021; Austin et al., 2021; Hendrycks et al., 2021; Lu et al., 2021; Li et al., 2022; Fried et al., 2023; Liu et al., 2023; Lai et al., 2023; Yu et al., 2024; Fu et al., 2023; Zheng et al., 2023) is the task of predicting subsequent code tokens based on the given context, benefits from datasets, which provide extensive code repositories from various programming languages. These datasets enable models to learn syntactic and semantic patterns (Jiao et al., 2023). Code editing (Just et al., 2014; Lin et al., 2017; Zhu et al., 2022b;a; Hu et al., 2023; Yan et al., 2023; Ahmad et al., 2023; Jiao et al., 2023; Zhang et al., 2023; Tian et al., 2024) involves automatically generating changes to existing code, such as bug fixes or refactoring. Datasets like EvalGPTFix (Zhang et al., 2023) and DebugBench (Tian et al., 2024), which focus on bug fixing and code refinement tasks, are instrumental in this area. To our knowledge, given the necessity and challenges in library evolution (Jiang et al., 2024), refer to the detailed comparison in Table 6 and Appendix C, the proposed dataset VersiCode is the first large-scale code generation dataset, covering both code completion and code editing. Refer to Appendix C for a comprehensive comparison among datasets.

**Third-party Library Evolution:** Third-party library code is continually updated due to bug fixes, code refactoring, and the addition of new features, making it a significant research topic in software engineering (Zhang et al., 2020; 2021; Dilhara et al., 2021; Liu et al., 2021; Wang et al., 2020; Vadlamani et al., 2021; Haryono et al., 2021). Studies by Zhang et al. (2020) show that Python APIs often evolve by adding, deleting, or modifying parameters. Further research by Zhang et al. (2021) notes frequent API changes, including parameter updates. Dilhara et al. (2021) reveal that developers adjust their use of machine learning libraries in response to updates, while Liu et al. (2021) and Dig & Johnson (2006) find that undocumented changes in Android and Java can cause errors. Research on API deprecation highlights issues with documentation and the quality of suggested alternatives (Wang et al., 2020; Vadlamani et al., 2021; Haryono et al., 2021; Brito et al., 2018), showing that improvement in library evolution does not necessarily translate to better suggestions for deprecated APIs. VersiCode, unlike traditional software engineering research, studies API version evolution from an LLM perspective, exploring its impact on model training, code generation, and evaluation.

## 8 CONCLUSION

In conclusion, our research underscores the need for updated benchmarks that capture the dynamic nature of software development, better assessing the capabilities of LLMs in code generation. By introducing the VersiCode dataset, we provide a realistic testing ground that reveals significant limitations in current models, like GPT-4o and LLaMA3, when handling version-specific code. Our findings advocate for continuous model improvements and the adoption of our new metric, i.e., critical diff check, which more accurately evaluates model performance against real-world challenges. This work not only introduces valuable tools but also sets a direction for future enhancements in AI-driven code generation, ensuring LLMs remain effective and relevant in professional settings. For future research, we will investigate a solution for version-controllable code generation based on the insights from this paper, including approaches like continual learning, memory-enhanced methods, or retrieval-based methods. Additionally, we plan to develop a live version of VersiCode, which will continuously incorporate new libraries and downstream use cases.

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

| Procedure | Rules |
|---|---|
| Ranked Libraries ↦ StackOverflow Q&A | Filter out answers that involve the use of libraries from the ranked libraries, and ensure these answers include content in the library version format (e.g., pandas==1.3.5) as well as code snippets. |
| Ranked Libraries ↦ Library Source Code | Based on the ranked libraries, parse the source code of these libraries to find functions related to version changes. |
| Ranked Libraries ↦ Downstream Application | (1)Exclude files that do not utilize libraries and version information explicitly listed in requirements.txt. (2)Exclude files with an average line length exceeding 100 characters. (3)Exclude files with a maximum line length exceeding 1000 characters. (4)Exclude files with less than 25% of alphabetic characters. (5)Exclude files with syntax errors. |
| Annotation ↦ Metadata | StackOverflow: Filter out data that has been annotated by experts with correct library version and code snippet, and utilize GPT to generate functionality descriptions for the code snippets. |
| | Library Source Code: Utilize GPT to extract examples from version change function docstrings, filter out successfully extracted data, and employ GPT to generate functionality descriptions for the examples. |
| | Downstream Application: Utilize GPT to generate functionality descriptions for code snippets. |

Table 3: Detailed explanation of annotation stages and the corresponding filtering rules.

# A  DATASET CONSTRUCTION

VersiCode is a large-scale code generation benchmark dataset focusing on evolving library dependencies. We propose two tasks to simulate real-world applications: version-specific code completion and version-aware code migration, incorporating version information into code generation constraints. First, we discuss data curation, and preprocessing of noisy code snippets and FAQs into organized metadata. Based on the metadata, we describe the task design and quality control process. We then address tagging API lifespan features per library version. Finally, we provide data statistics for VersiCode and discuss future dataset extensions.

## A.1  DATASET CURATION AND COLLECTION

As shown in Figure 7, we first collected permissively licensed Python repositories from GitHub that serve as the source code for Python libraries. These repositories are ranked by their popularity (as indicated by their collected stars). Using the list of popular libraries, we gathered data from three sources for each library: (1) Library Source Code: We collected all available versions of the library source code from GitHub, verifying with PyPI to ensure that the collected versions are formally released and can be installed via pip. From the library source code, we extracted official usage examples for each API from the docstrings. (2) Downstream Application Code: Given Python's popularity in scientific programming, we collected the source code from top-tier research papers over 10 years as downstream applications. These applications are valuable due to being lightweight yet self-consistent, diverse in their topics, and tagged release timelines associated with publishing venues. Given the time span, this data source implicitly includes evolving libraries. (3) Stack Overflow: Using the library names as queries, we collected FAQ data from Stack Overflow, which provides real user queries and diverse user answers. We filtered the data to include only those queries that explicitly mention the versions of the libraries used, using heuristic rules, as shown in Table 3. Additionally, we have made our best efforts to filter all of the source code based on the open-source licenses of the repositories to ensure there is no infringement.

Given the high diversity and varied quality of the collected raw data, we adopted a hybrid annotation approach involving both human experts and LLMs, such as ChatGPT. (1) Library Source Code: The library version is concrete and explicitly available, but example usage varies across libraries and versions. We used an LLM with in-context learning to help extract example code from docstrings, preparing the library version and code snippets. (2) Downstream Applications: The version can easily be extracted from configuration files, typically named "requirements.txt". We carefully filtered out Python files that are too long, do not mention the library version, or fail to compile. (3) Stack Overflow: Given the diversity of the questions, we designed strict heuristic rules to preliminarily annotate the library name, version, and corresponding Python code snippets mentioned in answers. We then distributed the pre-annotated data to six qualified human experts for verification and correction, ensuring the library version and code snippets are ready as well. With all pairs of library versions and code snippets, we employed ChatGPT with in-context learning to generate descriptions of the functionality for each code snippet. Each pair is wrapped in well-organized metadata.

Figure 7: The preprocessing pipeline to obtain metadata, structured as n-gram tuple of ⟨library name, version, functionality description, code snippet⟩.

## A.2 LIFECYCLE TAGGING OF APIS

Consider an API $a$ added to the library $L$ in version $V_s$ and deprecated in version $V_e$, and is active in the intermediate version $V_m$ where $s \leq m \leq e$. We refer to the interval $[s, e)$ as the *lifespan* of $a$. To analyze model performance in detail, we assessed how up-to-date each LLM was concerning newly added or deprecated APIs per version. We compared the source code between any two consecutive versions of each library to detect changes in API or method names. Based on the detection results, we labeled the datasets obtained from the library source code as follows: "addition" indicates an API newly added in the current version and still applicable in subsequent versions; "deprecation" indicates the current version is the last usable version for the API; and "general" indicates the API usage method is inherited from the previous version.

## A.3 DATA PREPARATION FOR EVALUATION

**Data Preparation for Token-level Code Completion.** As introduced in Section 2, we designed two types of version-controllable code generation tasks: version-specific code completion and version-aware code migration. The task granularities are categorized into token-level, line-level, and block-level to control difficulty and simulate different application scenarios. To better understand model performance, each instance in VersiCode is also tagged with the following: (1) Data source, which includes library source code, downstream applications, and Stack Overflow; (2) Feature type, including addition, deprecation, and general; (3) Release time, i.e. the timestamp from GitHub and Stack Overflow); These tags allow us to filter the evaluation dataset and gain sharper insights into model performance.

**Data Preparation for Execution-based Multi-granularity Code Completion.** As shown in Figure 5, we have constructed a subset for dynamic code analysis that includes executable test cases. From the data originating from library source code in VersiCode, we filter for data that includes complete context (e.g., import statements) code snippets. Experts interact with the web version of GPT-4 to refactor the code snippets into task functions. After a manual check of the task functions, experts interact with GPT-4 to write test cases for them. During the interaction, experts provide appropriate feedback to GPT-4. The test cases are run in a testing environment containing specific library versions (e.g., pandas==1.3.5); if successful, the annotation is completed after further manual verification, and if failed, more detailed feedback is provided to GPT-4 to assist with corrections. The annotated task function is processed into code completion forms with three levels of ⟨mask⟩ granularity: token, line, and block. The executable test cases include four types: (1) **Test return type**: tests whether the return type is correct. (2) **Test normal input**: tests whether the expected output is produced with normal inputs. (3) **Test boundary values**: tests whether special values (such as null values, incorrect types, etc.) are handled properly. (4) **Test functionality**: tests whether the function fulfills its primary functionality. The first three types of test cases have one instance per task function, while the fourth type has 1-3 instances.

**Data Preparation for Code Migration.** As shown in Figure 2, considering code migration instances constructed from pairs of metadata, the differences between source and target code versions result in

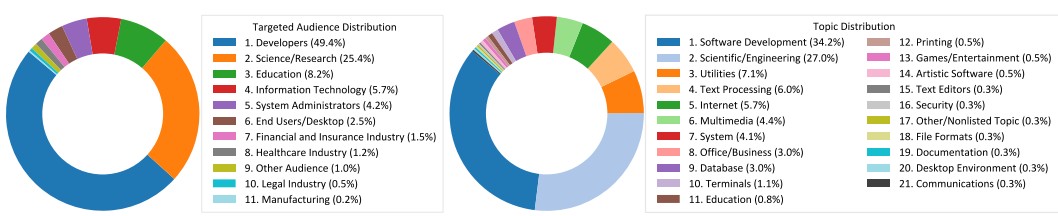

Figure 8: A proportional chart based on the classification system of targeted audience and topics in third-party Python libraries on PyPI.

various situations, such as updates from an older version to a newer version or vice versa. Additionally, we categorized versions according to version patterns, for example, treating torch v1.0.0 as a major version and torch v1.3.1 as a minor version, to identify combinations of major and minor version migration cases.

## B  DATA STATISTICS AND SCOPE

**Dataset Statistics**: We present the statistics of VersiCode in Table 4, using the StarCoder2's (Lozhkov et al., 2024) tokenizer to compute the number of tokens. We also outline the complete version of VersiCode in the table, which furnishes human-labeled data for three additional languages: C#, Java, and JavaScript. Our executable data, applied in Section 4, is a high-quality human-annotated subset from VersiCode, covering 12 libraries, 40 versions, and 119 functionality descriptions. For each functionality description, we matched 4 to 5 test cases.

| # Language | Python | | | | | Java | C# | JavaScript |
|---|---|---|---|---|---|---|---|---|
| # Data Source | StackOverflow; Library Source Code; Downstream Application | | | | | StackOverflow | StackOverflow | StackOverflow |
| # Num. of Libraries | 300 | | | | | 19 | 16 | 33 |
| # Num. of Versions | 2,207 | | | | | 25 | 16 | 60 |
| # Size of Meta Data | 11,268 | | | | | 29 | 16 | 62 |
| # Task Type | Completion | | | Editing (old to new) | Editing (new to old) | Completion | Completion | Completion |
| # Granularity | Token | Line | Block | Block | Block | Block | Block | Block |
| # Avg. Input Token | 2,087 | 2,075 | 55 | 191 | 195 | 57 | 63 | 67 |
| # Avg. Output Token | 2 | 16 | 128 | 131 | 128 | 349 | 255 | 167 |
| # Num. of Instances | 13,488 | 13,490 | 1,617 | 38,037 | 38,037 | 32 | 21 | 82 |

Table 4: Data statistics of VersiCode, including multiple languages.

**Scope**: VersiCode supports version-specific code completion at the token, line, and block levels, enabling developers to navigate through version variations effortlessly. It also facilitates block-level version-aware code editing, empowering users to make precise modifications tailored to requirements of each version. The collected metadata also serves as a valuable resource for potential customized task modifications, supported domains are illustrated in Figure 8, aiding in fine-tuning workflows and enhancing model training for optimal performance.

## C  RELATED DATASET

**Code Completion Datasets.** As shown in Table 5, we compare the VersiCode-completion dataset with existing benchmarks. VersiCode stands out in annotated data size, marking it as the inaugural dataset tailored for version-specific generation.

**Code Migration Datasets.** As shown in Table 6, we compare the VersiCode-migration dataset with existing benchmarks. VersiCode stands out in annotated data size, marking it the inaugural dataset tailored for version-specific migration.

| Benchmark | Source | Language | Samples | Completion Task | Granularity | Collection Time | Annotation |
|---|---|---|---|---|---|---|---|
| StaQC (Yao et al., 2018) | StackOverflow | Python, SQL | 267,056 | Function Programming | Line-Level, Block-Level | 2018 | None |
| CoNaLa (Yin et al., 2018) | StackOverflow | Python, Java | 2,879 | Function Programming | Line-Level, Block-Level | 2018 | Human |
| CT-maxmin (Feng et al., 2020) | Existing Benchmark | Multi(=6) | 2,615 | Cloze Test | Token-Level | 2020 | None |
| HumanEval (Chen et al., 2021) | Hand-Written | Python | 164 | Function Programming | Line-Level, Block-Level | 2021 | Human |
| MBPP (Austin et al., 2021) | Hand-Written | Python | 974 | Function Programming | Block-Level | 2021 | Human |
| APPS (Hendrycks et al., 2021) | Programming Sites | Python | 10,000 | Function Programming | Line-Level, Block-Level | 2021 | None |
| CT-all (Lu et al., 2021) | Existing Benchmark | Multi(=6) | 176,115 | Cloze Test | Token-Level | 2021 | None |
| CodeContests (Li et al., 2022) | Existing Benchmark, Codeforces | Multi(=3) | 13,610 | Function Programming | Block-Level | 2022 | None |
| HumanEval-FIM (Fried et al., 2023) | Existing Benchmark | Python | 164 | Function Programming | Line-Level, Block-Level | 2022 | None |
| HumanEval+ (Liu et al., 2023) | Existing Benchmark | Python | 164 | Function Programming | Line-Level, Block-Level | 2023 | LLM |
| MBPP+ (Liu et al., 2023) | Existing Benchmark | Python | 378 | Function Programming | Block-Level | 2023 | LLM |
| DS-1000 (Lai et al., 2023) | StackOverflow | Python | 1,000 | Function Programming | Line-Level, Block-Level | 2023 | Human |
| CoderEval (Yu et al., 2024) | Github | Python, Java | 460 | Function Programming | Block-Level | 2023 | Human |
| CodeApex (Fu et al., 2023) | Programming Sites | C++ | 476 | Function Programming | Block-Level | 2023 | None |
| HumanEval-X (Zheng et al., 2023) | Existing Benchmark | Multi(=5) | 820 | Function Programming | Line-Level, Block-Level | 2023 | Human |
| BigCodeBench (Zhuo et al., 2024) | Existing Benchmark | Python | 1,140 | Function Programming | Block-Level | 2024 | Human, LLM |
| **VersiCode** | StackOverflow, Github | Python, Java, C#, JavaScript | 28,595 | Cloze Test, Function Programming | Token-Level, Line-Level, Block-Level | 2024 | Human, LLM |

Table 5: Comparison of VersiCode and other code completion datasets. VersiCode is the largest annotated dataset, covering multiple languages and granularities, and involving both human and LLM joint annotations.

| Benchmark | Source | Language | Samples | Editing Task | Granularity | Collection Time | Annotation |
|---|---|---|---|---|---|---|---|
| Defects4J (Just et al., 2014) | Open Source Programs | Java | 357 | Debug | Block-Level | 2014 | None |
| QuixBugs (Lin et al., 2017) | Quixey Challenges | Python, Java | 40 | Debug | Line-Level | 2017 | Human |
| CoST (Zhu et al., 2022b) | GeeksForGeeks | Multi(=7) | 132,046 | Code Translation | Line-Level, Block-Level | 2022 | None |
| XLCoST (Zhu et al., 2022a) | GeeksForGeeks | Multi(=8) | 1,083,000 | Code Translation | Line-Level, Block-Level | 2022 | None |
| InstructCoder (Hu et al., 2023) | Github | Python | 114,000 | Code Refinement | Block-Level | 2023 | LLM |
| MultilingualTrans (Yan et al., 2023) | Programming Sites | Multi(=8) | 30,419 | Code Translation | Block-Level | 2023 | None |
| NicheTrans (Yan et al., 2023) | Programming Sites | Multi(>8) | 236,468 | Code Translation | Block-Level | 2023 | None |
| LLMTrans (Yan et al., 2023) | Hand-Written | Multi(=8) | 350 | Code Translation | Block-Level | 2023 | Human |
| Avatar (Ahmad et al., 2023) | Programming Sites | Python, Java | 62,520 | Code Translation | Block-Level | 2023 | None |
| G-TransEval (Jiao et al., 2023) | Existing benchmark, GeeksForGeeks | Multi(=5) | 400 | Code Translation | Token-Level, Block-Level | 2023 | Human |
| EvalGPTFix (Zhang et al., 2023) | AtCoder | Java | 151 | Debug | Block-Level | 2023 | Human |
| DebugBench (Tian et al., 2024) | LeetCode | Multi(=3) | 4,253 | Debug | Block-Level | 2024 | LLM |
| **VersiCode** | Github | Python | 76,074 | Version Adaptation | Block-Level | 2024 | LLM |

Table 6: Comparison between VersiCode and other code editing datasets, with VersiCode standing out as the largest annotated dataset specifically tailored for version adaptation.

# D    ADDITIONAL EXPERIMENTS AND DETAILS

## D.1    EXTENSIVE COMPARATIVE STUDY ON LARGE LANGUAGE MODELS

In addition to the model depicted in Figure 3, comprehensive and detailed evaluation results are presented in Table 7, encompassing 23 models and sorted by the release time of each model.

In addition to the model depicted in Figure 9, comprehensive and detailed evaluation results are presented in Table 7, encompassing 23 models and sorted by the release time of each model.

**Even token-level code completion is challenging.** We present the EM@1 results for token-level code completion on VersiCode, sorted by release time (highlighted in green, see Figure 3-a1). Compared to the Pass@1 results on HumanEval (blue) and MBPP (orange), all models perform significantly worse on VersiCode (green). This result indicates the difficulty in disambiguating and recalling version-specific library usage. It is important to note that larger and more recent models, such as GPT-4o (M13) and LLaMA3-70B (M12), demonstrate significantly superior performance compared to other models. (See Appendix H for the error analysis of GPT-4o.)

## D.2    MULTI-LANGUAGE ANALYSIS

As depicted in Table 8, we perform the primary multi-language experiments. Counter-intuitively, the performance of LLMs in Java, JavaScript, and C# surpasses that in Python. This anomaly might be attributed to potential data leakage from the Stack Overflow dataset.

## D.3    BLOCK-LEVEL CODE COMPLETION V.S. CODE MIGRATION

We use Python's built-in function *"compile()"* to compile the generated code snippets to check whether they are syntactically correct. Upon comparing "w/o grammar verification" and "w grammar

| Release Time | Model | HumanEval | HumanEval+ | MBPP | MBPP+ | VersiCode | | | |
|---|---|---|---|---|---|---|---|---|---|
| | | EM@1 | EM@1 | EM@1 | EM@1 | Library Source Code | Downstream Application | StackOverflow | Total |
| 2023.06.14 | WizardCoder-15B-V1.0 (Luo et al., 2024c) | 56.7 | 50.6 | 64.3 | 54.2 | 0.17 | 0 | 0.1 | 0.06 |
| 2023.06.14 | WizardCoder-Python-7B-V1.0 (Luo et al., 2024c) | 50.6 | 45.1 | 58.5 | 49.5 | 6.62 | 0.17 | 5.45 | 2.66 |
| 2023.07.18 | Llama-2-7B (Touvron et al., 2023) | 12.8 | - | 20.8 | - | 6.57 | 0.46 | 4.76 | 2.74 |
| 2023.07.18 | Llama-2-13B-Chat (Touvron et al., 2023) | 18.3 | - | 30.6 | - | 3.71 | 0.06 | 3.41 | 1.51 |
| 2023.08.25 | CodeLlama-7B-Instruct (Rozière et al., 2023) | 34.8 | - | 44.4 | - | 17.77 | 0.62 | 17.8 | 7.62 |
| 2023.08.25 | CodeLlama-13B-Instruct (Rozière et al., 2023) | 42.7 | - | 49.4 | - | 28.45 | 2.47 | 32.05 | 13.5 |
| 2023.08.28 | CodeLlama-7B-Python (Rozière et al., 2023) | 38.4 | - | 47.6 | - | 3.4 | 0.03 | 2.35 | 1.28 |
| 2023.10.29 | DeepSeek-Coder-6.7B-Instruct (Guo et al., 2024) | 74.4 | 71.3 | 74.9 | 65.6 | 3.83 | 0.15 | 4.34 | 1.71 |
| 2023.11.11 | Mistral-7B-Instruct-V0.2 (Jiang et al., 2023) | 42.1 | 36 | 44.7 | 37 | 13.96 | 1.85 | 20.33 | 7.54 |
| 2024.01.25 | DeepSeek-Coder-7B-Instruct-V1.5 (Guo et al., 2024) | 75.6 | 71.3 | 75.2 | 62.2 | 26.7 | 4.51 | 44.77 | 15.71 |
| 2024.01.25 | GPT-3.5-Turbo (OpenAI, 2023b) | 76.8 | 70.7 | 82.5 | 69.7 | 40.55 | 30.48 | 65.95 | 37.59 |
| 2024.02.27 | StarCoder2-7B (Lozhkov et al., 2024) | 35.4 | 29.9 | 55.4 | 45.6 | 12.21 | 0.32 | 13.02 | 5.27 |
| 2024.02.27 | StarCoder2-15B (Lozhkov et al., 2024) | 46.3 | 37.8 | 66.2 | 53.1 | 29.7 | 2.9 | 35.79 | 14.55 |
| 2024.04.09 | CodeGemma-7B-Instruct (CodeGemma Team et al., 2024) | 60.4 | 51.8 | 70.4 | 56.9 | 31.8 | 0.76 | 31.29 | 13.36 |
| 2024.04.09 | CodeGemma-7B (CodeGemma Team et al., 2024) | 44.5 | 41.5 | 65.1 | 52.4 | 29.61 | 1.12 | 34.01 | 13.28 |
| 2024.04.10 | aiXCoder-7B (aiXcoder team, 2024) | 54.9 | - | 66 | - | 17.51 | 1.09 | 26.3 | 8.83 |
| 2024.04.15 | aiXCoder-7B-Base (aiXcoder team, 2024) | 43.2 | - | 62.2 | - | 20.41 | 0.94 | 26.37 | 9.59 |
| 2024.04.15 | CodeQwen1.5-7B (Bai et al., 2023) | 51.8 | 45.7 | 73.5 | 60.8 | 11.61 | 0.12 | 7.58 | 4.33 |
| 2024.04.15 | CodeQwen1.5-7B-Chat (Bai et al., 2023) | 83.5 | 78.7 | 79.4 | 69 | 12.16 | 0.33 | 9.2 | 4.81 |
| 2024.04.18 | Llama-3-8B (Meta LlaMa team, 2024) | 35.5 | 29.3 | 64.4 | 51.6 | 17.18 | 0.24 | 20.69 | 7.57 |
| 2024.04.18 | Llama-3-8B-Instruct (Meta LlaMa team, 2024) | 61.6 | 56.7 | 70.1 | 59.3 | 20.79 | 3.67 | 34.08 | 12.23 |
| 2024.04.18 | Llama-3-70B-Chat (Meta LlaMa team, 2024) | 77.4 | 72 | 82.3 | 69 | 33.76 | 50.93 | 64.35 | 47.55 |
| 2024.05.13 | GPT-4o (OpenAI, 2024) | 85.4 | 81.7 | 85.7 | 73.3 | 58.37 | 72.98 | 87.21 | 70.44 |

Table 7: Full evaluation results of EM@1 on token-level code completion compared to related datasets and different data sources. The results for related datasets are collected from the online leaderboard of Evalplus (Liu et al., 2023).

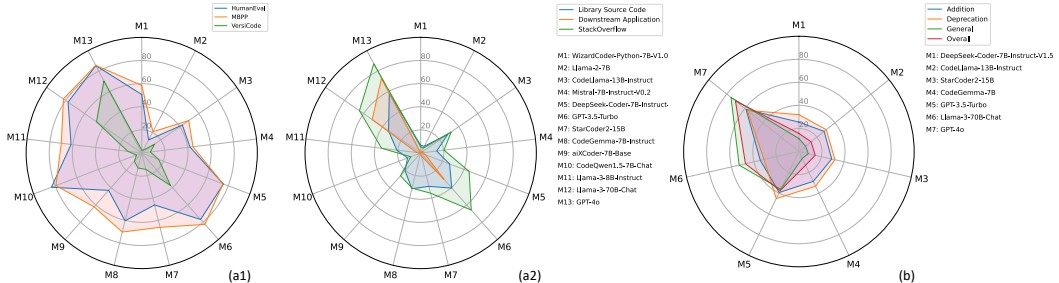

Figure 9: The *EM@1* results for token-level code completion from VersiCode: (a1) Comparison with existing benchmark datasets, (a) Performance grouped by data sources, and (b) Performance grouped by API lifecycle.

verification" in Table 9, it becomes evident that the model tasked with editing, alongside reference code snippets from other versions, finds it easier to produce grammar-verified code.

# E    METRIC DESIGN OF CRITICAL DIFF CHECK

## E.1    INTRODUCTION OF CRITICAL DIFF CHECK

Critical Diff Check (CDC) focuses on the changes in the code rather than the overall similarity of the entire code segment. CDC has five rules as follows:

- *Rule 1: Check whether the generated code contains the core token.*

- *Rule 2: Check whether the generated code is valid.*

- *Rule 3: Check if the number of arguments in the function using the core token is consistent.*

- *Rule 4: If the reference code uses a with statement, checks whether the generated code also uses a with statement.*

- *Rule 5: If the reference code uses keyword argument assignment, checks whether the generated code uses the same keyword argument assignment.*

The failure frequency and examples for each rule are shown in Table 10.

| Model | Python | | Java | | C# | | JavaScript | |
|---|---|---|---|---|---|---|---|---|
| | ISM@1 | PM@1 | ISM@1 | PM@1 | ISM@1 | PM@1 | ISM@1 | PM@1 |
| DeepSeek-Coder-7B-Instruct-V1.5 (Guo et al., 2024) | 40.03 | 27.35 | 61.55 | 46.62 | 71.43 | 49.68 | 75.22 | 54.24 |
| CodeLlama-13B-Instruct (Rozière et al., 2023) | 48.83 | 34.63 | 70.92 | 58.87 | 47.62 | 35.54 | 52.87 | 34.11 |
| StarCoder2-15B (Lozhkov et al., 2024) | 39.71 | 27.36 | 38.63 | 27.43 | 33.33 | 28.63 | 60.67 | 39.33 |
| CodeGemma-7B (CodeGemma Team et al., 2024) | 8.67 | 5.00 | 34.38 | 23.53 | 0 | 0 | 16.82 | 10.53 |
| GPT-3.5-Turbo (OpenAI, 2023b) | 40.77 | 28.06 | 50.00 | 39.34 | 28.57 | 26.87 | 24.39 | 15.85 |
| GPT-4o (OpenAI, 2024) | 64.72 | 50.48 | 70.83 | 64.04 | 71.43 | 63.26 | 77.74 | 70.24 |
| Llama-3-70B-Chat (Meta LlaMa team, 2024) | 57.68 | 41.47 | 61.55 | 58.57 | 66.67 | 56.35 | 75.61 | 67.61 |

Table 8: Multi-language performance on VersiCode

| Model | Code Completion | | Code Migration (Old ↦ New) | | Code Migration (New ↦ Old) | |
|---|---|---|---|---|---|---|
| | Block-level | | Block-level | | Block-level | |
| | ISM | PM | ISM | PM | ISM | PM |
| *w/o grammar verification* | | | | | | |
| DeepSeek-Coder-7B-Instruct-V1.5 (Guo et al., 2024) | 40.03 | 27.35 | 46.17 | 37.20 | 42.94 | 33.66 |
| CodeLlama-13B-Instruct (Rozière et al., 2023) | 48.83 | 34.63 | 41.74 | 32.37 | 41.41 | 30.01 |
| StarCoder2-15B (Lozhkov et al., 2024) | 39.71 | 27.36 | 40.94 | 30.73 | 44.46 | 31.88 |
| CodeGemma-7B (CodeGemma Team et al., 2024) | 8.67 | 5.00 | 24.54 | 17.46 | 22.61 | 12.08 |
| GPT-3.5-Turbo (OpenAI, 2023b) | 40.77 | 28.06 | 45.96 | 36.80 | 46.96 | 35.76 |
| Llama-3-70B-Chat (Meta LlaMa team, 2024) | 58.08 | 41.78 | 33.37 | 23.51 | 42.94 | 29.36 |
| GPT-4o (OpenAI, 2024) | 64.72 | 50.48 | 55.48 | 45.80 | 55.36 | 52.33 |
| *w grammar verification* | | | | | | |
| DeepSeek-Coder-7B-Instruct-V1.5 (Guo et al., 2024) | 0.00 | 0.00 | 45.41 | 36.44 | 40.25 | 28.89 |
| CodeLlama-13B-Instruct (Rozière et al., 2023) | 4.34 | 3.12 | 39.17 | 27.94 | 39.12 | 26.17 |
| StarCoder2-15B (Lozhkov et al., 2024) | 1.36 | 0.79 | 35.59 | 26.72 | 41.41 | 27.64 |
| CodeGemma-7B (CodeGemma Team et al., 2024) | 0.37 | 0.22 | 9.16 | 4.12 | 9.72 | 5.28 |
| GPT-3.5-Turbo (OpenAI, 2023b) | 40.28 | 27.57 | 45.96 | 36.80 | 46.96 | 35.06 |
| Llama-3-70B-Chat (Meta LlaMa team, 2024) | 64.73 | 50.48 | 54.72 | 45.04 | 55.36 | 52.33 |
| GPT-4o (OpenAI, 2024) | 57.68 | 41.47 | 33.37 | 23.51 | 42.94 | 29.36 |

Table 9: Results of block-level code completion and migration with or without grammar verification.

### E.2 ABLATION STUDY OF CRITICAL DIFF CHECK

We conducted ablation experiments on the five CDC rules and calculated the Pearson correlation coefficient with the Pass@1 metric for each, to demonstrate the reliability of CDC. The specific experimental data is shown in Table 11.

## F RUNNING EXAMPLE OF EXECUTABLE TEST

As shown in Figure 10, this is an example of a task function used for code generation, where the task function is processed in various granular forms of code completion. The "core token" is only provided for visualization, which is unseen for models. "library version" is optional, identified as "w/ or w/o version", and "import" statements are also optional, identified as "w/ or w/o import" in Table 1. As shown in Figure 11, these are the test cases for the task function illustrated in Figure 10. The test cases were developed by experts through interactions with GPT-4 and include four types of tests.

## G EVALUATION DETAILS

### G.1 HYPER-PARAMETER

As illustrated in Table 12, we have itemized the hyper-parameters pertinent to version-controllable code generation.

### G.2 PROMPT TEMPLATE

We introduce the prompt template for token-level, line-level, and block-level evaluations in Figure 12, Figure 13, and Figure 14, respectively.

```python
Task Function:

# Library Version: accelerate==0.16.0
# Core Token: release_memory

import torch
from torch import Tensor
from accelerate.utils import release_memory

def task_function(size: tuple) -> (Tensor, Tensor):
    """
    Creates two tensors filled with ones, processes them using an in-place memory release function,
    and returns them.

    Parameters:
    size (tuple): A tuple specifying the dimensions of the tensors to be created.

    Returns:
    tuple of torch.Tensor: A tuple containing two tensors, both located on the appropriate device
    (GPU if available, otherwise CPU).
    """
    device = 'cuda' if torch.cuda.is_available() else 'cpu'
    a = torch.ones(size, device=device)
    b = torch.ones(size, device=device)
    release_memory(a, b)
    return a, b
```

Figure 10: The ground truth for block-level code generation, used for Section 4. Note that, "core token" is only provided for visualization, which is unseen for models. "library version" is optional, identified as "w/ or w/o version", and "import" statements are also optional, identified as "w/ or w/o import" in Table 1.

```python
Test Cases:

import unittest
from unittest.mock import patch

class TestTaskFunction(unittest.TestCase):

    def test_return_type(self):
        """Test if the return type of the function is as expected (tuple of Tensors)."""
        result_a, result_b = task_function((1000, 1000))
        self.assertIsInstance(result_a, Tensor)
        self.assertIsInstance(result_b, Tensor)

    def test_normal_input(self):
        """Test the function with normal input and check if the results are as expected."""
        result_a, result_b = task_function((10, 10))
        self.assertEqual(result_a.size(), (10, 10))
        self.assertEqual(result_b.size(), (10, 10))

    def test_boundary_values(self):
        """Test the function with boundary values such as zero dimensions."""
        result_a, result_b = task_function((0, 0))
        self.assertEqual(result_a.numel(), 0)
        self.assertEqual(result_b.numel(), 0)

    @patch('__main__.release_memory')
    def test_functionality_1(self, mock_release_memory):
        """Test to verify if the release_memory function is called within the task_function."""
        task_function((50, 50))
        mock_release_memory.assert_called_once()

if __name__ == '__main__':
    unittest.main()
```

Figure 11: The test cases associated with generated code for dynamic code analysis, used for Section 4.

| | Model | Rule 1 | Rule 2 | Rule 3 | Rule 4 | Rule 5 |
|---|---|---|---|---|---|---|
| Token | GPT-3.5-Turbo | 363 (50.84%) | - | - | - | - |
| | LLaMA-3-70b-chat | 325 (45.52%) | - | - | - | - |
| | GPT-4o | 169 (23.67%) | - | - | - | - |
| Line | GPT-3.5-Turbo | 290 (40.62%) | 199 (27.87%) | 386 (54.06%) | 36 (5.04%) | 468 (65.55%) |
| | LLaMA-3-70b-chat | 258 (36.13%) | 124 (17.37%) | 332 (46.5%) | 36 (5.04%) | 478 (66.95%) |
| | GPT-4o | 150 (21.01%) | 67 (9.38%) | 229 (32.07%) | 7 (0.98%) | 390 (54.62%) |
| Block | GPT-3.5-Turbo | 320 (44.82%) | 3 (0.42%) | 443 (60.64%) | 31 (4.34%) | 489 (68.49%) |
| | LLaMA-3-70b-chat | 286 (40.05%) | 10 (1.4%) | 408 (57.14%) | 31 (4.34%) | 470 (65.83%) |
| | GPT-4o | 254 (35.57%) | 54 (7.56%) | 359 (50.28%) | 33 (4.62%) | 439 (61.48%) |
| Matching Rule | | $a \in c$ | `compile(c)` is successful | $|params_{o}(f)| = |params_c(f)|$ | $startwith(c', \text{'with'}) = startwith(c, \text{'with'})$ | $\forall p \in Kc'(f), p \in Kc(f)$ |
| Example | Core Token | wait_for_everyone | - | EvaluationSuite | clear_environment | init_on_device |
| | Positive Example | state.wait_for_everyone() | for i in range(5): print(i) | suite = EvaluationSuite.load("evaluate/evaluation-suite-ci") | with clear_environment(): | with init_on_device(device=device): |
| | Negative Example | state.wait_for_others() | for i in range(5) print(i) | suite = EvaluationSuite("imdb", "lvwerra/distilbert-imdb") | clear_environment() | init_on_device(layer, device) |

Table 10: Each rule of the CDC, along with the frequency, occurrence rate, and examples of mismatches for each rule. 'a' represents the core token, 'c' represents the code generated by the model, 'c′' represents the reference code, 'f' represents the function of the specified token, and 'params' refers to the function's parameter list. 'Kc′(f)' and 'Kc(f)' represent the keyword parameter lists of the reference code and the model-generated code, respectively, and 'p' represents the parameter assigned using keyword arguments. In detail, *Rule 1* checks whether the generated code contains the core token; *Rule 2* checks whether the generated code is valid; *Rule 3* checks if the number of arguments in the function using the core token is consistent; *Rule 4*, if the reference code uses a with statement, checks whether the generated code also uses a with statement; *Rule 5*, if the reference code uses keyword argument assignment, checks whether the generated code uses the same keyword argument assignment.

| Ablation | Model | CDC w/o Rule 1 | CDC w/o Rule 2 | CDC w/o Rule 3 | CDC w/o Rule 4 | CDC w/o Rule 5 | Pass@1 | CDC@1 |
|---|---|---|---|---|---|---|---|---|
| Token | GPT-3.5-Turbo | 49.16 | 49.16 | 49.16 | 49.16 | 49.16 | 41.88 | 49.16 |
| | LLaMA-3-70b-chat | 54.48 | 54.48 | 54.48 | 54.48 | 54.48 | 46.08 | 54.48 |
| | GPT-4o | 76.33 | 76.33 | 76.33 | 76.33 | 76.33 | 65.97 | 76.33 |
| Line | GPT-3.5-Turbo | 72.13 | 28.85 | 31.37 | 27.59 | 36.69 | 26.47 | 27.59 |
| | LLaMA-3-70b-chat | 82.63 | 29.27 | 31.09 | 29.27 | 49.3 | 32.07 | 29.27 |
| | GPT-4o | 90.62 | 41.32 | 44.68 | 41.04 | 64.99 | 46.08 | 41.04 |
| Block | GPT-3.5-Turbo | 99.58 | 26.75 | 30.39 | 26.75 | 38.37 | 11.48 | 26.61 |
| | LLaMA-3-70b-chat | 98.60 | 28.15 | 32.21 | 28.29 | 41.46 | 13.73 | 27.73 |
| | GPT-4o | 92.44 | 34.31 | 34.45 | 32.07 | 45.38 | 19.19 | 31.65 |
| *Pearson Correlation Coefficient with Pass@1* | | | | | | | | |
| | PCC | -0.5674 | 0.9069 | 0.9081 | 0.909 | 0.9029 | - | **0.9124** |

Table 11: Ablation study of Critical Diff Check per rule. The configuration labeled as "CDC w/o Rule i", where $i \in \{1, 2, 3, 4, 5\}$ means that when calculating the CDC score, Rule i is excluded, and only the other four rules are considered. The Pearson correlation coefficient calculates the correlation the metric's results obtained in each configuration against Pass@1.

## G.3 DATA SAMPLING

For token-level completion tasks(Figure 3), we randomly sampled 2,000 instances for evaluation. We used the entire executable dataset for line- and block-level completion tasks due to its smaller size (Figure 6, Table 1). In the time trend experiment (Figure 4), we sampled 200 data points per quarter or used all available data if fewer. And in the code migration task (Table 2), we randomly sampled 2,000 instances for evaluation.

## H ERROR ANALYSIS

### H.1 ERROR ANALYSIS OF GPT4-O

Despite GPT4-o achieving superior performance in general evaluation, it still encounters errors in 30% of instances. We provide several negative examples in Figure 16, Figure 17, and Figure 18.

| hyper-parameter | code completion | | | code migration |
|---|---|---|---|---|
| | token-level | line-level | block-level | block-level |
| temperature | 0.8 | 0.8 | 0.8 | 0.8 |
| top_p | 0.95 | 0.95 | 0.95 | 0.95 |
| max_tokens | 64 | 128 | 512 | 512 |
| n | 100 | 6 | 6 | 6 |

Table 12: Hyper-parameters for completion and migration.

```
prompt = f"""
You are a Python programming expert. Your task is to analyze a code snippet and infer the content masked by <token_mask>. Here
are your instructions:
1. You will receive:
   - A Python library name and its version, which is relevant to the content masked by <token_mask>
   - A code snippet with one or more <token_mask> markers

2. Each <token_mask> in the snippet represents the same masked content.

3. Based on the provided library and its version, infer the specific token that <token_mask> is hiding.

4. Provide your response as follows:
   - Give only ONE answer, regardless of how many <token_mask> appear
   - Include ONLY the inferred content
   - Wrap your answer with ```python and ``` to denote it as a code block
   - Omit any explanations or extra information

The Python library with its version and the code snippet are provided below:
Library and Version:
{dependency_version}

Code Snippet:
{masked_code}

Your response:
"""
```

Figure 12: Prompt template for token-level version-specific code completion.

```
prompt = f"""
You are a Python programming expert. Your task is to analyze a code snippet where a certain line is masked by <line_mask> and infer
the content of that line. Here are your instructions:
1. You will receive:
   - The name and version of the library relevant to this line of code
   - A code snippet with a <line_mask>

2. The <line_mask> represents a single masked line of code.

3. Based on the provided library information, infer what the <line_mask> is hiding.

4. Provide your response as follows:
   - Give only the inferred line of code
   - Wrap your answer with ```python and ``` to denote it as a code block
   - Omit any explanations or extra information

The code snippet and library information are provided below:
Libraries and Version:
{dependency_version}

Code Snippet:
{masked_code}

Your response:
"""
```

Figure 13: Prompt template for line-level version-specific code completion.

```
prompt = f"""
You are a professional Python engineer. Your task is to write Python code that implements a specific function based on the provided
library and version. Here are your instructions:
1. You will receive:
   - The name and version of the library relevant to the code
   - A code snippet with a <block_mask> where you need to infer the missing code

2. Based on the library information, write the Python code that fills the <block_mask> and implements the feature.

3. Provide your response as follows:
   - Return only the code that fills the <block_mask> and implements the function
   - Enclose your code with ```python and ``` to denote it as a Python code block
   - Omit any explanations or extra information

The library information and partially masked code snippet are provided below:
Library and Version:
{dependency_version}

Code Snippet with <block_mask>:
{masked_code}

Your response:
"""
```

Figure 14: Prompt template for block-level version-specific code completion.

```
prompt = f"""
You are now a professional Python programming engineer. I will provide you with a code snippet and a description of its functionality,
including the dependencies and versions used in the code. Then, I will provide the same dependencies but with a specified new version.
Your task is to refactor the code using the methods provided by the specified new version and return the refactored code. Please note
that you only need to return the refactored code and enclose it with <start> and <end>:
###Functionality description of the code
{description}
###Dependency and old version
{old_version}
###Old version code
{old_code}
###Dependency and new version
{new_version}
###Refactored new code
"""
```

Figure 15: Prompt template for version-aware code migration.

```
"dependency": "paddlepaddle-gpu",
"version": "==2.0.0",
"time": "2021-01-21",
"description": "The code creates a dataset using the PaddlePaddle framework and sets the variables to be used as data and labels for the dataset.",
"code": "import paddle\ndataset = paddle.distributed.fleet.DatasetBase()\ndataset._set_use_var([data, label])",
"masked_code": "import paddle\ndataset = paddle.distributed.fleet.DatasetBase()\ndataset.<mask>([data, label])",
"answer": "_set_use_var",
"type": "name_change_new",
"granularity": "token",
"language": "python",
"task": "code completion",
"source": "docstring",
"model_output_token_clear": "['set_use_var', 'set_filelist', 'set_list', 'set_filelist', 'set_filelist', 'set_data', 'set_filelist', 'set_list', 'set_list', 'set_filelist', 'set_list',
 'set_filelist', 'set_filelist', 'set_filelist', 'set_list', 'set_use_var', 'set_filelist', 'set_filelist', 'set_filelist', 'set_list', 'set_list', 'set_vars', 'set_filelist', 'set_list',
 'set_use_var', 'set_filelist', 'set_data', 'set_list', 'set_use_var', 'set_filelist', 'set_filelist', 'set_inputs', 'set_inputs', 'set_use_var', 'set_filelist', 'set_filelist', 'set_use_var',
 'set_filelist', 'set_data', 'set_list', 'set_filelist', 'set_use_vars', 'set_list', 'set_list', 'set_inputs', 'set_filelist', 'set_filelist', 'set_filelist', 'set_filelist', 'set_use_var', 'set_list',
 'set_use_var', 'set_filelist', 'set_list', 'set_list', 'set_list', 'set_filelist', 'set_records', 'set_inputs', 'set_use_var', 'set_filelist', 'set_filelist', 'set_filelist', 'set_list', 'set_use_var',
 'set_inputs', 'set_filelist', 'set_filelist', 'set_filelist', 'set_filelist', 'set_filelist', 'set_use_var', 'set_list', 'set_filelist', 'set_use_var', 'set_filelist', 'set_filelist', 'set_data',
 'set_filelist', 'set_list', 'set_inputs', 'set_list', 'set_filelist', 'set_vars', 'set_use_var', 'set_list', 'set_inputs', 'set_inputs', 'set_inputs', 'set_filelist', 'set_filelist', 'set_list',
 'set_filelist', 'set_filelist', 'set_filelist', 'set_filelist', 'set_filelist', 'set_list', 'set_data', 'set_use_vars']"
```

Figure 16: The first negative example of GPT-4o on token-level code completion.

```
"dependency": "pandas",
"version": "==1.1.0",
"time": "2020-07-28",
"description": "This code defines multiple aggregation functions with relabeling for the 'a' column, including 'max' and 'min'.",
"code": "is_multi_agg_with_relabel(a=\"max\")\nis_multi_agg_with_relabel(a_max=(\"a\", \"max\"), a_min=(\"a\", \"min\"))\nis_multi_agg_with_relabel()",
"masked_code": "<mask>(a=\"max\")\n<mask>(a_max=(\"a\", \"max\"), a_min=(\"a\", \"min\"))\n<mask>()",
"answer": "is_multi_agg_with_relabel",
"type": "name_change_new",
"granularity": "token",
"language": "python",
"task": "code completion",
"source": "docstring",
"model_output_token_clear": "['agg', 'agg', 'agg', 'agg', 'agg', 'agg', 'agg', 'agg', 'agg', 'agg', 'agg', 'agg', 'agg', 'agg', 'agg', 'agg', 'agg', 'agg', 'agg', 'agg',
 'agg', 'agg', 'agg', 'agg', 'agg', 'agg', 'agg', 'agg', 'agg', 'agg', 'agg', 'agg', 'agg', 'agg', 'agg', 'agg', 'agg', 'agg', 'agg', 'agg', 'agg', 'agg', 'agg', 'agg', 'agg', 'agg', 'agg', 'agg',
 'agg', 'agg', 'agg', 'agg', 'agg', 'agg', 'agg', 'agg', 'agg', 'agg', 'agg', 'agg', 'agg', 'agg', 'agg', 'agg', 'agg', 'agg', 'agg', 'agg', 'agg', 'agg', 'agg', 'agg', 'agg', 'agg', 'agg',
 'agg', 'agg', 'agg', 'agg', 'agg', 'agg', 'agg', 'agg', 'agg', 'agg', 'agg', 'agg', 'agg', 'agg', 'agg', 'agg', 'agg', 'agg', 'agg', 'agg', 'agg', 'agg']"
```

Figure 17: The second negative example of GPT-4o on token-level code completion.

"dependency": "wandb",
"version": "==0.12.12",
"time": "2022-04-05",
"description": "The code loads the latest version of a previously logged model with the name \"my-simple-model\" and assigns the model object to the variable \"model\".",
"code": "# assuming you have previously logged a model with the name \"my-simple-model\"\nsm = use_model(\"my-simple-model:latest\")\nmodel = sm.model_obj()\n",
"masked_code": "# assuming you have previously logged a model with the name \"my-simple-model\"\nsm = <mask>(\"my-simple-model:latest\")\nmodel = sm.model_obj()\n",
"answer": "use_model",
"type": "add",
"granularity": "token",
"language": "python",
"task": "code completion",
"source": "docstring",
"model_output_token_clear": "['load_model', 'Artifact.get', 'use_artifact', 'use_artifact', 'load_model', 'Artifact', 'load_model', 'Artifact.get',
 'load_model', 'load_model', 'load_model', 'load_model', 'load_model', 'load_model', 'load_model', 'use_artifact', 'Artifact', 'Artifact', 'Artifact',
 'Artifact.load', 'load_model', 'load_model', 'load_model', 'load_model', 'Artifact.get', 'load_model', 'use_artifact', 'load_model',
 'load_model', 'load_model', 'load_model', 'use_artifact', 'load_model', 'load_model', 'load_model', 'load_model', 'load_model', 'load_model',
 'load_model', 'Artifact', 'load_model', 'load_model', 'load_model', 'load_model', 'load_model', 'load_model', 'load_model', 'load_model',
 'use_artifact', 'use_artifact', 'use_artifact', 'use_artifact', 'load_model', 'use_artifact', 'load_model', 'load_model', 'load_model', 'load_model', 'load_model',
 'use_artifact', 'load_model', 'load_model', 'load_model', 'load_model', 'use_artifact', 'load_model', 'load_model', 'load_model', 'load_model', 'load_model',
 'load_model', 'load_model', 'load_model', 'load_model', 'use_artifact', 'load_model', 'load_model', 'load_model', 'use_artifact', 'load_model',
 'use_artifact', 'Artifact', 'Artifact', 'use_artifact', 'use_artifact', 'load_model', 'Artifact', 'use_artifact', 'Artifact', 'load_model', 'Artifact', 'load_model',
 'load_model', 'load_model', 'use_artifact', 'load_model', 'use_artifact', 'use_artifact', 'load_model']"

Figure 18: The third negative example of GPT-4o on token-level code completion.