# OpenReview forum: "VersiCode: Towards Version-controllable Code Generation"
_ICLR.cc/2025/Conference — Submitted to ICLR 2025_

### Official Review · Reviewer_sSYw · 2024-11-04

**Soundness:** 2
**Presentation:** 2
**Contribution:** 2
**Rating:** 3
**Confidence:** 4

**Summary:**

This work introduces a benchmark for measuring the ability of LLMs trained on code to condition their predictions on API versions. It measures both code completion and code migration performance of a wide range of models across deprecated, newly added, and consistently available API methods across a large set of projects in a new benchmark called VersiCode. It also introduces a new accuracy metric based on static analysis. Results show that more capable LLMs generally fair worse on newly added and deprecated methods than on ones that have consistently been present, while smaller models show the reverse pattern. All models show a decrease in performance for newer APIs. Most models struggle especially with code migration across major version changes.

**Strengths:**

This work provides a dataset that fills an important gap in code evaluations specifically, measuring the effect of API changes. The dataset it contributes is quite large and contains useful data from a variety of sources. It provides some of the first large-scale empirical evidence that LLMs perform differently on certain types of APIs, including more recent releases and deprecated ones. It presents a wide range of results.

**Weaknesses:**

My main concerns are around the interpretation (and significance) of the results, the clarity and soundness of the methodology, and the relevance of the contributions to this venue. I'll address these in order, followed by a list of minor issues.

**Results:** the interpretation of the results involves questionable claims in a number of places. Some of these are perhaps mostly clarity issues, while a few others seem more foundational.

An example of the latter is the comparison with HumanEval and MBPP in Fig. 3, which shows, in Fig. 3.a1, that completion accuracy is far lower on VersiCode than on the aforementioned benchmarks. The text argues that this shows "the difficulty in disambiguating and recalling version-specific library usage" (L209)  and that LLMs "struggle to deliver satisfactory results" (L232). However, self-contained functions are not a good basis of comparison for examples of real API usage. It is well-established that completing real-world code, especially function calls, is harder than code in self-contained function with detailed specifications. That is not to mention the extensive leakage concerns around the baseline, benchmark programs.

This work doesn't offer a baseline of version-agnostic API completion to compare the results on version-sensitive API completion with. The closest is the comparison in Fig. 3.b, which compares the completion quality on methods that are deprecated/added vs. ones that stay the same ("general") in the particular version used in a sample's prompt. Since unchanging APIs make up more than half the total volume of API calls (Fig. 4), it stands to reason that many of these are consistent across most/all versions of their libraries, making the "general" performance a rough approximation of the performance of models on API call completion disregarding versions. The gap between these and version-sensitive API calls is considerably smaller for the large models in this subfigure, and even inverted for the smaller ones (the explanation for which, on L248, is a bit unsatisfying as it would seem to apply to large models equally). That is, version-sensitive API calls are only slightly harder to complete (for large models) than ones that stay the same.

This combined with Fig. 4's trends suggests that the "real" effect is mostly one of frequency: older APIs and ones that do not get deprecated are used more often in typical training sets, which tend to go back to GitHub's inception, and are therefore more accurately completed. Whether there is an actual difference in performance between these three types of methods is questionable. Figure 4 shows almost indistinguishable curves for each model over time between the subfigures, although separating out "general" might show a more clear contrast. This raises strong concerns around whether there is a difference at all, or whether the differences observed in Fig. 3.b are actually caused by confounds related to statistical frequency over time.

It is also not quite clear why performance on deprecated methods is measured by prompting with the version right before the method is being deprecated. At least, that is implied by L134 ("the last usable version") -- L129 contradicts itself by referring to the lifespan of a method as both $s \leq m \leq e$ and $[s, e)$, making it unclear whether version $e$ is included. If the prompt asks to complete a method that is about to be deprecated in the next version, I assume the goal is to find cases where the models might incorrectly assume that the method is already deprecated? If so, are the samples constructed so that there is a new, valid completion in the exact same context in the subsequent version? If not, the models almost "have no choice" but to complete the soon-to-be-deprecated version for snippets that use an idiomatic style that is made obsolete by the deprecation. Similarly, was there a check that the models aren't simpy using an alternative, valid API? Replacement APIs are often introduced before deprecating older versions.

There are a number of other, mostly minor, questionable claims about the performance as well, including:

L242: why does "excelling in handling downstream applications [...] increase the likelihood of models memorizing specific content"? Is the causal statement meant to be reversed, viz. models memorizing the content increases the likelihood of excelling in downstream applications? That would make more sense, but it would need some empirical evidence to back up that: the smaller models are not memorizing content, the patterns included here show up in the training data, and larger models aren't simply better at pattern matching.

L249: "However, most models struggle with reasoning and adapting to intermediate versions." -- does "intermediate" refer to added/deprecated ones here? If so, Fig. 3b shows that "most models" do relatively better on those than on unchanged ones.

L416: the description of GPT-4o's performance here seems off: its major->major performance is worse than on any other task. Relatedly, it was not clear to me what the sentence on L417/418 meant. Also, the use of "major->minor" and similar syntax is a bit confusing. Typically, we only distinguish between a major or minor version change. This paper seems to use the word "major" to refer just to the first version of a major release in order to distinguish between cases such as the first version of one major change to another vs. jumping from a given minor change within a major release to the start of the next major release. It is unclear why performance on the latter is so much higher than on major->major changes, or even on minor->minor changes, which ought to be much easier.


**Methodology:** as evidenced by some of the questions/concerns above (e.g. the lack of a solid baseline, the definition of deprecation), there are multiple concerns with the details of the approach. These often stem from a lack of clarity in the text, including in the appendix. An example of this is the definition of code migration: the example in Fig. 2 shows a case where the same description applies to two subsequent API versions and the code differs by just the API version. However, the way descriptions are generated (L863) doesn't guarantee that equivalent code snippets in successive versions would get the exact same description. Were the migration pairs collected from cases that happened to have an identical description? Was any manual validation applied to this process to capture the rate of false positives (and ideally FNs, but those would be harder to measure)?

The concern around the lack of manual validation applies to a number of other methodological decisions, such as scraping code from scientific papers. While a convenient dataset for this task, scientific code has a less than stellar reputation for quality. Was any filtering applied to confirm that this code was not using deprecated methods? Many of the results rely on the evaluation samples being unambiguous, in the sense that there is exactly one valid completion, but there is little evidence in the methodology or results that confirms that this has been guaranteed by this benchmark.

The definition of the "CDC" metric also raises some concerns. It appears that "exact match" as used in this paper does not refer to a true (semantic) match between programs, but rather to an identifier ("core token", L1025) match. The CDC metric expands on this with a few additional heuristic checks, such as comparing the number of arguments (but not their exact values) and the presence of a `with` statement. Per Tab. 11, these added heuristics have an almost imperceptible (and perhaps not statistically significant) impact on the correlation between CDC and pass@1 ranking. The main evidence for its usefulness comes from it yielding values closer to those from pass@1 than EM in Tab. 1, at the line and block level and the more noticeably higher Pearson correlation compared to EM at the block level. While that is useful, the lower values may well be in part due to the introduction of false negatives (e.g. matching keyword arguments is not necessarily a prerequisite for correctness) and there is still a considerable gap at the block level. Overall, I find it questionable to define "exact match" as something quite far from a true, semantic match, and the additional heuristics in CDC clearly leave a sizable gap with the actual target metric. I would strongly encourage the benchmark to focus on execution correctness over heuristic match, even if that shrinks the set of usable examples substantially.

**Relevance:** on a slightly less pressing note, this work seems only partially relevant to this venue. While the dataset it contributes shows useful limitations of LLMs, it does not offer proven insights into where these limitations originate and how they can be addressed. It makes a number of claims about why these limitations exist, some of which I identify as dubious above, but it does not support these claims with explorations of the models' training data or learned representations. Without a clear relation between how the models are actually built and the observed effects, any explanations feel rather speculative. That leaves the main contribution in the domain of code evaluation data, rather than representation learning. It may be more suitable for a venue dedicated to code-specific learning.

**Minor Issues:**

- L93: nit: inverted final quote
- L143: no need to use double equals
- L412: please add a mention that Tab. 2 has the results.
- L431: missing "more"?

**Questions:**

Regarding the concerns around the (interpretation of the) results, I strongly suggest removing the comparison with HumanEval and MBPP in favor of an analysis that separates out API completion performance on methods that are (almost) never changed vs. ones that are, to better isolate the existence and magnitude of the effect of versioning on API completion specifically. On this note, please try to answer:

Is the difference in performance statistically significant when controlling for time? E.g. in Fig. 4, is there a consistent difference when comparing datapoints for the same year between APIs that don't change (ideally ones that never change, but "general" is fine too) and ones that are added/deprecated for any of these models? Reverse trends for addition vs. deprecation are interesting too.

Please address the questions in the discussion about the definition and evaluation of deprecated methods and on the other methodological concerns above.

What degree of manual validation was used to ensure that the underlying snippets are valid? How were code migration pairs constructed exactly?

Is there a need for the CDC metric compared to pass@1 if the latter can be applied to a sufficiently large set of samples already? How do you address the concerns around the metric's potential for false negatives and the relatively slim difference compared to "Exact Match" (and the concerns with that nomenclature)?

Answers to all other questions in the "Weaknesses" section would be appreciated.

---

> ### Author Response · Authors · 2024-12-03
> **Response to Reviewer sSYw (Part1)**
>
> Thank you for your detailed assessment of our paper. Your comments have significantly enhanced the quality of our work, especially in the analysis of our results.
>
> Below are our responses to your questions:
>
> ---
>
> 1. **"I strongly suggest removing the comparison with HumanEval and MBPP in favor of an analysis that separates out API completion performance on methods that are (almost) never changed vs. ones that are, to better isolate the existence and magnitude of the effect of versioning on API completion specifically."**
>
>    Following your suggestion, we have moved the comparison with HumanEval and MBPP from the main body to the appendix. While we acknowledge that these benchmarks may suffer from potential data contamination, we believe the comparison still provides readers with a baseline understanding of the challenges involved in version-sensitive API-calling code generation compared to general code generation.
>
> ---
>
> 2. **"Is the difference in performance statistically significant when controlling for time? E.g., in Fig. 4, is there a consistent difference when comparing data points for the same year between APIs that don't change (ideally ones that never change, but 'general' is fine too) and ones that are added/deprecated for any of these models? Reverse trends for addition vs. deprecation are interesting too."**
>
>    We categorized the "general API" data by year and obtained the following experimental results. Unlike the downward trends observed for addition and deprecation in Figure 4, performance on the general category remained relatively stable over time, with some models even showing improvement.
>
> | Instance Num                    | 72    | 60    | 586   | 369   | 379   | 596   | 350   |
> |---------------------------------|-------|-------|-------|-------|-------|-------|-------|
> | General                         | Y2017 | Y2018 | Y2019 | Y2020 | Y2021 | Y2022 | Y2023 |
> | DeepSeek-Coder-7B-Instruct-V1.5 | 1.46  | 4.07  | 19.65 | 24.51 | 23.62 | 12.1  | 21.85 |
> | CodeLlama-13B-Instruct          | 0.1   | 2.77  | 14.29 | 17.91 | 14.37 | 7.76  | 14.68 |
> | StarCoder2-15B                  | 4.28  | 1.25  | 15.63 | 19.97 | 16.61 | 9.04  | 18.76 |
> | CodeGemma-7B                    | 3.75  | 3.67  | 12.44 | 17.36 | 12.55 | 7.39  | 15.28 |
> | GPT-3.5-Turbo                   | 27.51 | 39.03 | 46.66 | 55.95 | 47.64 | 35.68 | 47.1  |
> | GPT-4o                          | 68.93 | 79.32 | 76.27 | 81.37 | 78.47 | 79.13 | 81.32 |
> | Llama-3-70B-Chat                | 47.61 | 59.28 | 55.67 | 60.38 | 58.37 | 51.29 | 60.89 |

---

> ### Author Response · Authors · 2024-12-03
> **Response to Reviewer sSYw (Part2)**
>
> 3. **"Please address the questions in the discussion about the definition and evaluation of deprecated methods and on the other methodological concerns above."**
>
>    We have revised the notes in Table 2 to clarify the setup for code migration:
>    *"The arrow $\mapsto$ indicates the direction of migration, where 'Major' denotes a major release (e.g., Torch v2.0.0 and v2.4.0), and 'Minor' denotes a minor release (e.g., Torch v2.1.3 and v2.3.4). Migration can be categorized as (1) `XX$\mapsto$Major`, crossing any major release (e.g., v2.0.0 to v2.4.0), and (2) `XX$\mapsto$Minor`, migrating within the same major release (e.g., v2.0.0 to v2.0.3). The 'Old $\mapsto$ New' scenario simulates upgrading to a newer version, while 'New $\mapsto$ Old' represents maintaining historical code."*
>
>    For code migration dataset construction, we focused solely on the "name change" type of API modifications, where the code differs only in core tokens, while the docstring descriptions remain unchanged. Consequently, code snippets from old and new versions share the same functional description.
>
> ---
>
> 4. **"What degree of manual validation was used to ensure that the underlying snippets are valid? How were code migration pairs constructed exactly?"**
>
>    For the executable sample set, we manually validated each sample by installing the corresponding library version in a Docker environment and executing the ground truth script to verify its correctness.
>
> ---
>
> 5. **"Is there a need for the CDC metric compared to pass@1 if the latter can be applied to a sufficiently large set of samples already? How do you address the concerns around the metric's potential for false negatives and the relatively slim difference compared to 'Exact Match' (and the concerns with that nomenclature)?"**
>
>    The CDC metric was introduced as a trade-off between evaluation accuracy and cost. For the VersiCode dataset, constructing a sufficiently large executable set would entail prohibitive costs due to several real-world constraints, including:
>    1. High annotation and verification costs for test cases.
>    2. Challenges in configuring development environments, such as ensuring compatibility between hardware (e.g., GPU versions) and system settings.
>    3. Significant upfront preparation costs, which hinder the dataset's broader adoption.
>
>    In contrast, the CDC framework, once refined and validated on the evaluation set, allows us to achieve relatively accurate and interpretable results at lower cost. Therefore, even with the availability of large datasets, the CDC metric remains valuable and necessary.

---

### Official Review · Reviewer_4QQX · 2024-11-05

**Soundness:** 3
**Presentation:** 2
**Contribution:** 3
**Rating:** 6
**Confidence:** 4

**Summary:**

The authors find that code generated by LLMs needs to target specific library versions, and this problem was ignored in previous work. They propose two new tasks, version-specific code completion, and version-aware code migration, to evaluate LLMs’ ability in version-controllable code generation. They construct a Python dataset, VersiCode, to evaluate LLMs on these two tasks. They also propose a new evaluation metric, Critical Diff Check (CDC), to assess code migration results. Evaluation results show that version-controllable code generation is a significant challenge, even for extremely powerful LLMs.

Contributions:

- They propose two novel important tasks, i.e. version-specific code completion and version-aware code migration.

- They construct a versioned dataset, VersiCode.

- They introduce a new metric, Critical Diff Check.

- Their experiments provide valuable insights and directions for future research.

**Strengths:**

Originality: The problem of version-controllable code generation is pretty novel. Previous research in code generation ignores that the libraries and APIs used by the generated code can evolve.

Quality: The paper constructs the first versioned code generation dataset, including version-specific code completion and version-aware code migration tasks. The authors evaluate many popular LLMs on the dataset and give valuable insights about their ability to generate version-controlled code.

Clarity: The paper clearly states the problem of version-controllable code generation, including the motivation and how existing benchmarks fail to address the challenge.

Significance: Version-controllable code generation is very important in practice. The generated code is used with specific library versions, so the code generation process must consider the library versions.

**Weaknesses:**

1. The only metric used in the version-aware code migration is CDC, a metric just proposed in this paper. Although in section 4, the CDC@1 is stated to align closely with pass@1, it is hard to say this new metric can serve as the only metric in code migration.

**Questions:**

1. Is it possible to use more familiar and possibly more reliable metrics, like pass@1, to evaluate code migration results?

2. The paper uses “Exact Match” to mean the specified API is used in the generated code. This practice is inconsistent with other papers, where “exact match” means the generated code is the same as the oracle. I hope the authors can change the name of this metric to a more appropriate one.

**Details Of Ethics Concerns:**

The authors must ensure that the open-source data does not contain sensitive data, such as personal privacy information.

---

> ### Author Response · Authors · 2024-12-03
> **Response to Reviewer 4QQX**
>
> ### Response to Reviewer 4QQX
>
> **Regarding the comment:**
> *"The only metric used in the version-aware code migration is CDC, a metric just proposed in this paper. Although in section 4, the CDC@1 is stated to align closely with pass@1, it is hard to say this new metric can serve as the only metric in code migration."*
>
> We appreciate your valuable feedback. During the rebuttal period, we conducted intensive data annotation and experimental evaluations to address this concern. Following a process similar to executable test case annotation for code completion, we annotated and evaluated the performance of the CDC metric specifically for the code migration task. The experimental results are as follows:
>
> **New-to-Old Migration**
> | Model            | Pass@1 | EM@1   | ISM@1  | PM@1   | CDC@1  |
> |-------------------|--------|--------|--------|--------|--------|
> | GPT-3.5-Turbo     | 0.41   | 2.85   | 51.87  | 38.14  | 0      |
> | LLaMA-3-70B-Chat  | 4.88   | 10.16  | 55.02  | 47.13  | 1.63   |
> | GPT-4o            | 7.72   | 12.2   | 55.35  | 47.18  | 2.44   |
>
> **Old-to-New Migration**
> | Model            | Pass@1 | EM@1   | ISM@1  | PM@1   | CDC@1  |
> |-------------------|--------|--------|--------|--------|--------|
> | GPT-3.5-Turbo     | 7.32   | 27.64  | 67.1   | 52.03  | 1.22   |
> | LLaMA-3-70B-Chat  | 14.23  | 22.36  | 60.98  | 46.5   | 2.03   |
> | GPT-4o            | 34.15  | 63.41  | 80.89  | 66.07  | 16.26  |
> | PCC               |       -  |0.431  | 0.3203 | 0.5232 | 0.6016 |
>
> The results indicate that, although the evaluation accuracy of CDC shows a slight decline in the context of code migration compared to code completion tasks, CDC remains the metric with the highest consistency with pass@1 when compared to other static analysis metrics.
>
> ---
>
> **Regarding the comment:**
> *"The paper uses 'Exact Match' to mean the specified API is used in the generated code. This practice is inconsistent with other papers, where 'exact match' means the generated code is the same as the oracle. I hope the authors can change the name of this metric to a more appropriate one."*
>
> To avoid any potential ambiguity, we will revise the term *Exact Match* to *Core Token Match* in the revised version of the paper. We sincerely thank you for pointing this out and for your constructive suggestion.

---

### Official Review · Reviewer_dNit · 2024-11-08

**Soundness:** 4
**Presentation:** 3
**Contribution:** 3
**Rating:** 6
**Confidence:** 3

**Summary:**

The VersiCode paper addresses the crucial gap in current LLM capabilities by focusing on version-controllable code generation, a requirement in real-world software development due to frequent updates in third-party libraries. The authors introduce two novel tasks, version-specific code completion (VSCC) and version-aware code migration (VACM), to evaluate models on handling evolving software dependencies. A new dataset, VersiCode, consisting of over 300 Python libraries and 2,000 versions, enables comprehensive testing. They also propose the Critical Diff Check (CDC) metric, which enhances traditional metrics by evaluating version-specific aspects such as API usage and parameter handling. Through testing on models like GPT-4 and LLaMA, the study reveals significant challenges in version-sensitive code generation, underscoring the need for targeted pretraining, continual learning, and refined evaluation methods to improve LLM adaptability to evolving software versions.

**Strengths:**

Major Strengths

1. Comprehensive Dataset and Benchmark for Version-Specific Tasks: VersiCode is an important contribution as it directly addresses the under-explored area of version-specific code generation. By including metadata such as function descriptions, code snippets, and version numbers, it enables realistic evaluations of LLM capabilities in scenarios that require adherence to specific library versions. As shown in Figure 2, VersiCode’s metadata is utilized to create multi-granularity completion tasks, demonstrating the paper’s methodological depth in representing version-sensitive contexts.

2. Introduction of the CDC Metric: The Critical Diff Check (CDC) metric provides a more detailed assessment of LLM performance by focusing on crucial factors like API usage and parameter handling. This is particularly useful for evaluating tasks where syntax alone does not suffice. Table 1 presents a comparison of CDC and other metrics across different test cases, demonstrating its close alignment with the Pass@1 score, which highlights CDC’s effectiveness as a reliable proxy for assessing version-specific correctness.

3. Insightful Evaluation on Diverse Models and Detailed Error Analysis: The authors evaluate a wide range of models, including GPT-4o and StarCoder, revealing notable performance differences across version migration scenarios. For instance, in Table 2, models are evaluated across major and minor version changes, highlighting the difficulties that LLMs face in backward compatibility. This comprehensive analysis provides valuable insights into model limitations and suggests paths for improvement.

Minor Strengths

1. Detailed Task Decomposition with Multi-Level Completion: The dataset’s structure supports token, line, and block-level completion, making VersiCode versatile and relevant for different development use cases. Figure 3 compares model performance on token-level code completion across various data sources, showcasing the dataset’s capability to test LLM adaptability to different levels of code.

2. Rigorous Evaluation Methodology: The experimental setup is clearly defined, with the authors selecting 2,000 test instances for token-level completion (Section 3.1). This careful setup enhances the credibility of the results, providing a consistent baseline against well-known datasets like HumanEval[1] and MBPP[2].

[1] Chen M, Tworek J, Jun H, et al. Evaluating large language models trained on code[J]. arXiv preprint arXiv:2107.03374, 2021.

[2] Austin J, Odena A, Nye M, et al. Program synthesis with large language models[J]. arXiv preprint arXiv:2108.07732, 2021.

**Weaknesses:**

Major Weaknesses

1. Lack of Implementation of Retrieval-Augmented Generation (RAG): Although the authors acknowledge the potential of RAG techniques, there is no exploration of its integration, which could have significantly enhanced LLM performance in this context. RAG could assist in real-time access to documentation or version-specific information, potentially improving accuracy on challenging cases (see Section 5.2 for model limitations in handling context). Incorporating recent works in RAG-based methodologies for code tasks, such as those explored in [1], could inspire potential enhancements.

2. Absence of Evaluation Metrics for High-Level Reasoning: While CDC is effective for syntactic and structural validation, it doesn’t capture the semantic reasoning required to fully assess code functionality in context. Including metrics that evaluate semantic coherence or logical consistency could provide a more rounded view of LLM limitations, similar to the approaches suggested in studies like [2] which focus on code understanding and error detection.

Minor Weaknesses

1. Potential Bias in Model Evaluation Sampling: Although VersiCode’s dataset is extensive, the paper does not clarify how test samples were selected, which could impact the generalizability of findings. Describing the selection process, such as prioritizing recent or frequently updated APIs, would improve the robustness of conclusions (Section 2).

2. Limited Generalization to Languages Beyond Python: VersiCode primarily targets Python libraries, despite the widespread use of other languages in development. Expanding the dataset to include JavaScript or Java, for example, could enhance the benchmark’s applicability across diverse development environments, as is common in industry setting.

[1] Li R, Allal L B, Zi Y, et al. Starcoder: may the source be with you![J]. arXiv preprint arXiv:2305.06161, 2023.

[2] Le H, Wang Y, Gotmare A D, et al. Coderl: Mastering code generation through pretrained models and deep reinforcement learning[J]. Advances in Neural Information Processing Systems, 2022, 35: 21314-21328.

**Questions:**

The paper makes a significant contribution to understanding the performance of LLMs on version-specific code tasks, yet certain design choices would benefit from further clarification. The following questions aim to probe areas where additional context could strengthen the work:

1. Could the authors elaborate on the rationale behind the specific criteria used for sampling instances in model evaluations?

2. Did the team consider incorporating any RAG-based methods in future work to enhance the model’s access to relevant version documentation?

3. Are there plans to extend VersiCode to support other programming languages, given the diverse needs in software development?

---

> ### Author Response · Authors · 2024-12-03
> **Response to Reviewer dNit (Part1)**
>
> To Reviewer dNit,
>
> Thank you for your insightful comments. We have addressed your concerns as follows:
>
> 1. **"Lack of Implementation of Retrieval-Augmented Generation (RAG)"**
>
>    We have conducted additional experiments to explore Retrieval-Augmented Generation (RAG). Specifically, we implemented a retrieval mechanism based on embedding similarity. The knowledge base for retrieval includes open-source code from downstream applications, source code of third-party libraries (including their docstrings), and Q&A data from the StackOverflow community. We analyzed the experimental results from two perspectives: the sources of retrieval data and their relevance to the API lifecycle.
>
> | DataSource               | Downstream Code      | LibrarySourceCode     | Stackoverflow       |
> |--------------------------|----------------------|-----------------------|---------------------|
> | Pass@1 with/without rag  | 53.935 / 45.320     | 10.560 / 11.559       | 63.03 / 64.81       |
> | Pass@3 with/without rag  | 56.253 / 48.878     | 13.254 / 12.222       | 67.40 / 69.889      |
> | Pass@10 with/without rag | 58.634 / 51.902     | 13.995 / 12.726       | 70.45 / 73.084      |
>
> | Lifecycle Features        | General                 | Add                 | Deprecation         |
> |--------------------------|---------------------|---------------------|---------------------|
> | Pass@1 with/without RAG  | 37.61 / 39.33      | 36.269 / 35.89      | 27.31 / 43.12       |
> | Pass@10 with/without RAG | 42.868 / 48.46     | 44.462 / 42.972     | 34.973 / 50.618     |
>
>
> ---
>
>
>
> 2. **"Could the authors elaborate on the rationale behind the specific criteria used for sampling instances in model evaluations?"**
>
>    We employed a fully random sampling strategy aligned with the distribution of the complete dataset. This ensures the reliability of our evaluation results by maintaining representativeness across the dataset.
>
> ---
>
> 3. **"Are there plans to extend VersiCode to support other programming languages, given the diverse needs in software development?"**
>
>    The automated updating and maintenance of the VersiCode dataset is an ongoing effort, comprising three key components:
>    1. **Automated Data Collection:** We screen libraries from the PyPI classification system by selecting topics with stable update frequencies and libraries that exceed specific thresholds for GitHub stars or recent activity rates.
>    2. **Knowledge Extraction and Management:** For each library version, we construct API structural knowledge graphs and map API evolution relationships across versions.
>    3. **Dataset Generation and Maintenance:** Based on the evolution paths and distributions within the knowledge graphs, we select functionality points to generate evaluation samples.
>
>    Furthermore, we have extended VersiCode to include annotations for 771 evaluation samples from Java, C#, and JavaScript. For future work, we are following the Python pipeline to collect and analyze data for other programming languages, including Rust, Go, Swift, and Java. This effort already covers over 800 libraries and 7,000 versions of data.

---

> ### Author Response · Authors · 2024-12-03
> **Response to Reviewer dNit (Part2)**
>
> ---
>
> 4. **"Absence of Evaluation Metrics for High-Level Reasoning"**
>
>    This study focuses on assessing LLMs' ability to correctly invoke APIs of specified versions. The evaluation metric, CDC, is designed to align with this objective. To address your suggestion, we have provided supplementary results in Table 1 using the CodeBLEU metric. CodeBLEU incorporates longer context through data flow graphs and control flow graphs. Our results demonstrate that CDC outperforms CodeBLEU in evaluating VersiCode, confirming its suitability for our goals.
>
>
> | Model                                       | Pass@1 | EM@1   | ISM@1  | PM@1   | CDC@1 | codebleu | ngram_match_score | weighted_ngram_match_score | syntax_match_score | dataflow_match_score |
> |---------------------------------------------|--------|--------|--------|--------|-------|----------|-------------------|----------------------------|--------------------|----------------------|
> | w/ import; w/ version                       |        |        |        |        |       |          |                   |                            |                    |                      |
> | GPT-3.5-Turbo                               | 11.48  | 54.9   | 78.15  | 47.04  | 26.61 | 69.78    | 80.88             | 82.52                      | 65.8               | 67.79                |
> | Llama-3-70B-Chat                            | 13.73  | 59.1   | 79.1   | 48.02  | 27.73 | 72.48    | 81.12             | 83.72                      | 68.29              | 71.7                 |
> | GPT-4o                                      | 19.19  | 59.24  | 76.68  | 59.43  | 31.65 | 76.27    | 79.9              | 84.61                      | 73.26              | 76.29                |
> | w/o import; w/ version                      |        |        |        |        |       |          |                   |                            |                    |                      |
> | GPT-3.5-Turbo                               | 2.52   | 25.77  | 61.57  | 35.43  | 10.5  | 51.25    | 62.94             | 64.34                      | 48.03              | 48.28                |
> | Llama-3-70B-Chat                            | 3.81   | 34.03  | 68.12  | 43.91  | 10.78 | 60.43    | 62.38             | 69.39                      | 58.8               | 59.34                |
> | GPT-4o                                      | 6.72   | 45.52  | 74.85  | 51.91  | 18.07 | 65.98    | 62.82             | 71.88                      | 65.83              | 65.46                |
> | w/ import; w/o version                      |        |        |        |        |       |          |                   |                            |                    |                      |
> | GPT-3.5-Turbo                               | 12.61  | 57.14  | 78.42  | 48.21  | 28.71 | 50.39    | 63.3              | 63.75                      | 47.55              | 46.66                |
> | Llama-3-70B-Chat                            | 11.48  | 59.38  | 77.96  | 56.08  | 26.33 | 53.3     | 64.58             | 64.95                      | 49.97              | 50.88                |
> | GPT-4o                                      | 21.43  | 64.71  | 79.59  | 49.38  | 33.33 | 56.46    | 65.78             | 66.05                      | 55.36              | 52.84                |
> | w/o import; w/o version                     |        |        |        |        |       |          |                   |                            |                    |                      |
> | GPT-3.5-Turbo                               | 2.8    | 27.31  | 62.34  | 36.62  | 10.64 | 50.61    | 62.83             | 63.91                      | 46.92              | 47.92                |
> | Llama-3-70B-Chat                            | 3.22   | 33.47  | 66.52  | 42.52  | 12.61 | 60.67    | 61.05             | 69.7                       | 58.51              | 60.48                |
> | GPT-4o                                      | 4.76   | 46.36  | 74.95  | 50.15  | 17.93 | 65.51    | 65.42             | 71.6                       | 65.36              | 64.17                |
> | Pearson Correlation Coefficient with Pass@1 |        |        |        |        |       |          |                   |                            |                    |                      |
> | PCC                                         | -      | 0.9031 | 0.7977 | 0.6766 | 0.959 | 0.3414   | 0.5674            | 0.3888                     | 0.3161             | 0.2875               |
>
>    We acknowledge the need for future evaluations of LLMs in more complex software engineering tasks. CDC offers a framework for evaluation metric design that can be upgraded by refining its rule-based systems to accommodate high-level reasoning tasks.
>
> ---
>
> We hope these clarifications address your concerns. Thank you again for your valuable feedback!

---

### Official Review · Reviewer_orNc · 2024-11-08

**Soundness:** 3
**Presentation:** 3
**Contribution:** 3
**Rating:** 5
**Confidence:** 4

**Summary:**

The paper introduces VersiCode, a benchmark aimed at addressing the challenges faced by LLMs in generating version-specific code generation. It proposes two tasks: version-specific code completion (VSCC) and version-aware code migration (VACM). These tasks simulate real-world scenarios where deprecated API invocations must be updated to a newer version. The study includes a novel metric, Critical Diff Check (CDC@1), to access the code generation's adherence to version-specific requirements.

**Strengths:**

- Strong Insight. Capturing version dynamics is essential for effectively handling version-specific dependencies in practical software development.
- Comprehensive dataset. VersiCode covers 300 libraries over nine years, offering extensive coverage for version-specific testing and paving the way for future research in version-specific code generation and automated code migration.
- Novel metric. The CDC@k metric employs a set of hand-crafted rules to assess the similarity between API usages across different versions, enabling a more detailed, execution-free evaluation of version-specific code generation.

**Weaknesses:**

- Lack of in-depth analysis. As noted in subsection 4.2, the performance of GPT-4o drops significantly without import statements, suggesting that the evaluation is heavily influenced by prompt design and provided context. A more thorough analysis on how context information, such as documentation related to added or deprecated APIs, affects performance would be beneficial.
- Misleading analysis. In subsection 5.2, the claim that "The context code in another version is still helpful, but its benefits are limited" is difficult to corroborate based on Table 1 and Table 2, as block-level code migration details are not sufficiently presented. Additionally, the statement, "There is a significant improvement across most models, except for LLaMA3-70B and GPT-4o, detailed in Appendix D.3," is puzzling, as Appendix D.3 focuses on grammar verification impacts. Clarification is needed to confirm if there was an error or oversight.

**Questions:**

See above.

---

> ### Author Response · Authors · 2024-12-03
> **Response to Reviewer orNc**
>
> Thank you for your valuable comments.
>
> ---
>
> **1. In response to:**
> *"A more thorough analysis on how context information, such as documentation related to added or deprecated APIs, affects performance would be beneficial."*
>
> Thank you for your insightful feedback regarding subsection 4.2. Indeed, providing elements like "import" statements or pre-loading related API documentation essentially supplies the model with additional contextual references when generating API call code. This, as you noted, can impact the accuracy of evaluating the model's internal knowledge. It's also why the performance of the "with import" condition surpasses that of the "without import" condition.
>
> To address your suggestion, we conducted an additional set of experiments involving retrieval-augmented generation. In these experiments, we employed an embedding-based retrieval mechanism. The associated knowledge base included open-source code from downstream applications, source code of third-party libraries (including docstrings), and corpus data from Stack Overflow Q&A discussions. We analyzed the impact from two perspectives: (1) the data sources for retrieval, and (2) the API lifecycle (e.g., added or deprecated APIs). The results and corresponding discussions have been included to provide a more comprehensive analysis.
>
> | DataSource               | Downstream Code      | LibrarySourceCode     | Stackoverflow       |
> |--------------------------|----------------------|-----------------------|---------------------|
> | Pass@1 with/without rag  | 53.935 / 45.320     | 10.560 / 11.559       | 63.03 / 64.81       |
> | Pass@3 with/without rag  | 56.253 / 48.878     | 13.254 / 12.222       | 67.40 / 69.889      |
> | Pass@10 with/without rag | 58.634 / 51.902     | 13.995 / 12.726       | 70.45 / 73.084      |
>
>
> | Lifecycle Features        | General                 | Add                 | Deprecation         |
> |--------------------------|---------------------|---------------------|---------------------|
> | Pass@1 with/without RAG  | 37.61 / 39.33      | 36.269 / 35.89      | 27.31 / 43.12       |
> | Pass@10 with/without RAG | 42.868 / 48.46     | 44.462 / 42.972     | 34.973 / 50.618     |
>
> ---
>
> **2. In response to:**
> *"Misleading analysis."*
>
> We apologize for any confusion caused by our previous wording and have refined the statements in subsection 5.2 for better clarity. Specifically:
>
> In Tables 1 and 2, we presented analysis results for block-level code completion and block-level code migration, respectively. In Appendix D.3, we further compared the experimental results of these two code generation tasks under the "without import" and "with version" settings, as shown in Table 9.
>
> Moreover, we evaluated the impact of grammar verification under both "with grammar verification" and "without grammar verification" conditions. By comparing the results in Table 9, we concluded that while contextual code from another version is still beneficial, its advantages are limited.
>
> We acknowledge that the original title of Appendix D.3, "Grammar Verification Impact," may have been misleading. In the revised version, we have renamed Appendix D.3 to **"Block-Level Code Completion vs. Code Migration"** to better reflect its content and avoid misinterpretation.
>
> ---
>
> Thank you again for your constructive comments, which have significantly improved the clarity and rigor of our analysis.

---

### Official Review · Reviewer_hhqh · 2024-11-11

**Soundness:** 3
**Presentation:** 3
**Contribution:** 2
**Rating:** 6
**Confidence:** 4

**Summary:**

Despite current LLMs' remarkable advances in code generation tasks, they are typically trained and evaluated using static benchmarks such as HumanEval and MBPP. However, real-world software development is dynamic, with frequent library updates and API changes that require version-specific code compatibility. Current benchmarks do not address these challenges, highlighting a gap between LLM evaluations and practical development needs. To bridge this gap, the paper proposes two new tasks: (i) version-specific code completion (VSCC) and version-aware code migration (VACM)—and introduce VersiCode, a comprehensive dataset covering over 300 Python libraries and 2,000 versions. Additionally, the authors present a new metric, Critical Diff Check (CDC),  to better assess LLMs' ability to handle version-specific code generation.

**Strengths:**

- The paper addresses a real-world software development challenge, which is largely ignored by LLM4Code literature, where frequent updates to libraries and APIs require code to be compatible with specific versions.

- It introduces two new tasks focused on code evolution: code completion and version-aware code migration. These tasks are common in real-world development, and the paper provides detailed, separate approaches for each.

- The paper presents VersiCode, a large, high-quality dataset that includes data from over 300 Python libraries across over 2,000 versions over nine years.

- The new CDC metric improves on standard code similarity checks by including essential details like API usage, parameter handling, and managing deprecated features.

**Weaknesses:**

- While dealing with API evolution, the paper did not deal with more complex changes involving updates of API parameters or behaviors over time.

- The proposed VersiCode dataset requires regular updates to remain relevant, and handling thousands of library versions could become difficult as libraries continue to evolve, which may limit long-term usefulness.

**Questions:**

Your work provides valuable advancements for version-aware code generation. However, handling complex API changes—such as alterations in method behaviors, parameter types, or the introduction of new functionalities—can significantly impact code compatibility and functionality. How do you plan to address these more intricate API modifications in future iterations of VersiCode and CDC?

Additionally, what strategies are you considering to ensure that VersiCode remains up-to-date and scalable as libraries continue to evolve rapidly across different programming languages and ecosystems?

---

> ### Author Response · Authors · 2024-12-03
> **Response to Reviewer hhqh**
>
> Thank you for your insightful questions, which align closely with our ongoing research efforts surrounding VersiCode. Below are detailed responses to your queries:
>
> ---
>
> **1. How do you plan to address more intricate API modifications in future iterations of VersiCode and CDC?**
>
> As highlighted in Table 2 of [1], while there are 14 fundamental types of API evolution in Python, **Method Removal** (28.3%) and **Method Addition** (29.2%) collectively account for 57.5% of all changes. These types form the primary focus of VersiCode's current scope. In future work, we aim to extend our considerations to include **Field Removal** (16.7%) and **Field Addition** (17.4%), as well as **Class Removal** (2.3%) and **Class Addition** (2.4%). Additionally, we will explore combinations of these fundamental evolution types through data mining during dataset construction. For instance, **Method Renaming** could be modeled as a combination of **Method Removal** followed by **Method Addition**, and might also involve **Parameter Addition** or **Parameter Reordering**.
>
> Regarding the CDC evaluation framework, we aim to emphasize two practical principles:
> 1. Rule-based systems or decision trees inherently offer strong interpretability and low computational overhead for code evaluation.
> 2. Code evaluation differs from natural language evaluation—each token (e.g., spaces, indentation, identifiers, operators) distinctly impacts correctness. Instead of solely measuring the overall similarity between code and ground truth, we should focus on critical tokens that may cause runtime errors.
>
> The current CDC metrics are tailored to the VersiCode dataset, emphasizing the accuracy of API call evaluations. Future enhancements to the CDC may include:
> - Incorporating **data flow** and **control flow** analysis, enabling API-related code slicing.
> - Introducing decision trees to address potential conflicts or redundancies in the rules.
> - Leveraging distillation mechanisms to automate adjustments to the rule-based system, adapting to different evaluation perspectives across diverse code benchmarks.
>
> ---
>
> **2. What strategies are you considering to ensure VersiCode remains up-to-date and scalable as libraries evolve rapidly across programming languages and ecosystems?**
>
> Automated updating and maintenance of the VersiCode dataset are ongoing efforts, involving three key strategies:
> 1. **Automated Data Collection**: For Python, we leverage the PyPI [2] classification system to identify libraries with stable update frequencies and significant GitHub stars or rapid growth in star count. These libraries form the basis of API structural knowledge graphs across versions.
> 2. **Automated Knowledge Extraction and Management**: We construct evolution knowledge graphs that capture API changes between versions. These graphs are then used to identify functional points for generating evaluation samples.
> 3. **Knowledge-Based Dataset Generation and Maintenance**: Evaluation samples are created based on evolutionary paths and distributions derived from the knowledge graphs.
>
> For multi-language support, the current VersiCode dataset includes 771 evaluation samples from Java, C#, and JavaScript. Moving forward, we are extending the VersiCode construction workflow to additional languages such as Rust, Go, Swift, and Java. This involves collecting source code and downstream applications for over 800 libraries spanning more than 7,000 versions.
>
> ---
>
> **References:**
> [1] Zhang, Zhaoxu, et al. "How do Python framework APIs evolve? An exploratory study." 2020 IEEE 27th International Conference on Software Analysis, Evolution and Reengineering (SANER). IEEE, 2020.
> [2] [PyPI](https://pypi.org/search/)

---

### Meta-Review · Area_Chair_7dBp · 2024-12-22

**Metareview:**

This paper addresses the issue that code generated by LLMs needs to target specific library versions, a problem overlooked in previous work. While the reviewers found the topic interesting and recognized its potential, several concerns were raised. Several expert reviewers have identify that the interpretation and significance of the results need further clarification, and the methodology requires more transparency and soundness to be fully convincing. Additionally, several technical issues need to be addressed. The authors are encouraged to refine the presentation, clarify the methodology, and strengthen the technical details. With these improvements, the work could make a valuable contribution in future submissions.

**Additional Comments On Reviewer Discussion:**

While the reviewers found the topic interesting and recognized its potential, several concerns were raised. Several expert reviewers have identify that the interpretation and significance of the results need further clarification, and the methodology requires more transparency and soundness to be fully convincing.

---

### Decision · Program_Chairs · 2025-01-22

Reject